# Decentralized Accelerated Proximal Gradient Descent

**Haishan Ye**[1] *      **Ziang Zhou**[1] *      **Luo Luo**[2]      **Tong Zhang**[2]

[1]Shenzhen Research Institute of Big Data, The Chinese University of Hong Kong, Shenzhen

[2]Department of Mathematics, The Hong Kong University of Science and Technology

hsye_cs@outlook.com      zhouza15@fudan.edu.cn      luoluo@ust.hk      tongzhang@ust.hk

## Abstract

Decentralized optimization has wide applications in machine learning, signal processing, and control. In this paper, we study the decentralized composite optimization problem with a non-smooth regularization term. Many proximal gradient based decentralized algorithms have been proposed in the past. However, these algorithms do not achieve near optimal computational complexity and communication complexity. In this paper, we propose a new method which establishes the optimal computational complexity and a near optimal communication complexity. Our empirical study shows that the proposed algorithm outperforms existing state-of-the-art algorithms.

## 1   Introduction

In this paper, we consider the decentralized composite optimization problem where agents aim to solve the following composite convex problem defined as

$$\min_{x \in \mathbb{R}^d} h(x) \triangleq f(x) + r(x), \quad f(x) \triangleq \frac{1}{m} \sum_{i=1}^{m} f_i(x), \tag{1}$$

where each function $f_i(x)$ is the loss function of agent $i$ and known only to the agent; $r(x)$ is a non-smooth, convex function shared by all agents. The agents form a connected and undirected network. Agents can communicate with their neighbors to cooperatively solve the Problem (1).

Many machine learning problems can be formulated as Problem (1) such as the elastic net [29], the graphical Lasso [4], and sparse logistic regression [24]. Because centralized optimization suffers from the communication traffic jam on the central server and robustness of the network, the decentralized optimization has become an active research topic in machine learning. Decentralized methods for solving Problem (1) have also been widely studied in signal processing, control, and optimization communities.

Due to the wide applications, many different decentralized algorithms have been proposed in the past. We review the existing methods for the smooth non-proximal case where $r(x) = 0$. Some earlier primal algorithms were penalty based, and they can only reach sublinear convergence $\mathcal{O}(1/t)$ even for strongly convex objective functions [12, 27, 7]. The methods in [20] established the linear convergence for decentralized methods based on ADMM. More recently, gradient tracking based algorithms were proposed and they all achieved linear convergence including EXTRA [18], ESOM [11] , exact diffusion [28], NIDS [9], and many others [14, 15, 3, 17, 8]. Qu & Li [15] combined acceleration technique with gradient tracking to obtain faster convergence rate. Some other recent works [16, 22, 5, 5] studied the dual formulations of the problem, and proposed accelerated dual gradient descent to reach an optimal convergence rate for smooth problems. Recently, Ye et al. [26] proposed `Mudag`, which can achieve the optimal computation complexity and near optimal communication complexity for the smooth case with $r(x) = 0$.

---

Many gradient tracking based algorithms have been extended to decentralized composite optimization problems with a non-smooth regularization term such as `PG-EXTRA` [19] and `NIDS` [9]. However, due to the non-smooth term, these algorithms can only achieve sub-linear convergence rates. Recently, the authors of [21] proposed a gradient tracking based method called `SONATA`, and established a linear convergence rate with the assumption that $f(x)$ is strongly convex. In addition, Alghunaim et al. [2] proposed a primal-dual algorithm which can achieve linear convergence rate when each $f_i(x)$ is convex. Recently, a unified framework to analyze a large group of algorithms, and showed that these algorithms can also achieve linear convergence rates with nonsmooth regularization term such as `EXTRA (PG-EXTRA)` [18], `NIDS` [9], and `Harnessing` [14] in the work of [1, 25]. Despite intensive studies in the literature, the convergence rates of these previous algorithms do not match the optimal convergence rate. Moreover, the communication complexities achieved by algorithms analyzed in the framework of Xu et al. [25] and Alghunaim et al. [1] are sub-optimal.

In this paper, we propose a novel algorithm that can establish the optimal computation complexity and a near optimal communication complexity. Our method is closely related to `Mudag` of [26], which can only handle the non-composite case with $r(x) = 0$. We summarize our contributions as follows:

1. We proposed a novel decentralized primal proximal algorithm that can achieve the optimal computational complexity $\mathcal{O}(\sqrt{\kappa_g} \log(\frac{1}{\epsilon}))$ and an almost optimal communication complexity $\mathcal{O}\left(\sqrt{\frac{\kappa_g}{1-\lambda_2(W)}} \log(\frac{p(M,L,\mu)}{q(M,L,\mu)}) \log \frac{1}{\epsilon}\right)$, which matches the lower bound up to a $\log$ factor. $\kappa_g$ is the global condition number of $f(x)$. $M$ and $L$ are the smoothness parameter of local $f_i(x)$ and $f(x)$, respectively. $p$ and $q$ are polynomials of $M$, $L$ and $\mu$ of order no larger than three. To the best of our knowledge, our work is the first near optimal decentralized proximal algorithm.

2. Our algorithm does not require each individual function to be (strongly) convex. Thus, our algorithm has a wide application range since $f_i(x)$ may not be convex in many machine learning problems. The strong convexity condition of each $f_i(x)$ is required in the algorithms suited in the framework of [25] to achieve linear convergence rates. `SONATA` can also achieve linear convergence rate without this condition [21]. However, `SONATA` has to construct successive convex approximation function to deal with the non-convexity of $f_i(x)$ which requires extra computation burden.

## 2 Problem Set-up

We first introduce some properties of objective functions that will be used in this paper. Then we introduce the Gossip matrix associated with the network. Finally, we reformulate the Problem (1).

### 2.1 Function Properties

**Global $L$-Smoothness**    We call $f(x)$ is $L$-smooth, that is, for any $y, x \in \mathbb{R}^d$, it holds that

$$f(y) \leq f(x) + \langle \nabla f(x), y - x \rangle + \frac{L}{2} \|y - x\|^2.$$

**Global $\mu$-Strong Convexity**    We call $f(x)$ is $\mu$-strongly convex, that is, for any $y, x \in \mathbb{R}^d$, it holds that

$$f(y) \geq f(x) + \langle \nabla f(x), y - x \rangle + \frac{\mu}{2} \|y - x\|^2.$$

**Local $M$-Smoothness**    For each $f_i(x)$ in Eqn. (1), and any $y, x \in \mathbb{R}^d$, it holds that

$$f_i(y) \leq f_i(x) + \langle \nabla f_i(x), y - x \rangle + \frac{M}{2} \|y - x\|^2.$$

**Local $\nu$-Strong Convexity**    For each $f_i(x)$ in Eqn. (1), and any $y, x \in \mathbb{R}^d$, it holds that

$$f_i(y) \geq f_i(x) + \langle \nabla f_i(x), y - x \rangle + \frac{\nu}{2} \|y - x\|^2.$$

Based on the smoothness and strong convexity, we can define global and local condition number of the objective function respectively as follows

$$\kappa_g = L/\mu \quad \text{and} \quad \kappa_\ell = M/\nu.$$

It is well known that $\kappa_g \leq \kappa_\ell$ and $L \leq M$. Furthermore, if $f(x)$ is $\mu$-strongly convex and $r$ is convex, then one can easily check that $h(x) = f(x) + r(x)$ is also $\mu$-strongly convex.

## 2.2 Gossip Matrix

We use $W \in \mathbb{R}^{m \times m}$ to denote the gossip matrix associated with the network and use $\lambda_2(W)$ to denote the second largest eigenvalue of $W$. The gossip matrix $W$ has the following important properties:

1. $W$ is symmetric with $W_{i,j} \neq 0$ if and only if agents $i$ and $j$ are connected or $i = j$.
2. $\mathbf{0} \preceq W \preceq I$, $W\mathbf{1} = \mathbf{1}$, $\text{null}(I - W) = \text{span}(\mathbf{1})$.

We use $I$ to denote the $m \times m$ identity matrix and $\mathbf{1} = [1, \ldots, 1]^\top \in \mathbb{R}^m$ denotes the vector with all ones.

By above two properties, one can achieve averaging $x_i$'s in different agents by several steps of local communication (by multiplying $W$). Instead of directly multiplying $W$ several times, Liu & Morse [10] proposed a more efficient way to achieve averaging described in Algorithm 2 which has the following important proposition.

**Proposition 1.** *Let $\mathbf{x}^K$ be the output of Algorithm 2 and $\bar{x} = \frac{1}{m}\mathbf{1}^\top\mathbf{x}^0$. Then it holds that*

$$\bar{x} = \frac{1}{m}\mathbf{1}^\top\mathbf{x}^K, \quad \text{and} \quad \left\|\mathbf{x}^K - \mathbf{1}\bar{x}\right\| \leq \left(1 - \sqrt{1 - \lambda_2(W)}\right)^K \left\|\mathbf{x}^0 - \mathbf{1}\bar{x}\right\|,$$

*where $\lambda_2(W)$ is the second largest eigenvalue of $W$.*

## 2.3 Problem Reformulation

Denote by $x_i \in \mathbb{R}^d$ the local copy of the variable of $x$ for agent $i$. We introduce the aggregated variable $\mathbf{x}$ and the aggregated objective function $H(\mathbf{x})$ as

$$H(\mathbf{x}) = \frac{1}{m}\sum_{i=1}^m f_i(x_i) + \frac{1}{m}\sum_{i=1}^m r(x_i) \text{ with } \mathbf{x} = [x_1, \cdots, x_m]^\top. \tag{2}$$

We can reformulate Problem (1) as

$$\min_{\mathbf{x} \in \mathbb{R}^{m \times d}} H(\mathbf{x}) \quad \text{subject to} \quad x_1 = x_2 = \cdots = x_m. \tag{3}$$

We denote the aggregated smooth component, the aggregated non-smooth component, and $\nabla F(\mathbf{x})$ as

$$F(\mathbf{x}) = \frac{1}{m}\sum_{i=1}^m f_i(x_i), \ R(\mathbf{x}) = \frac{1}{m}\sum_{i=1}^m r(x_i), \ \nabla F(\mathbf{x}) = \frac{1}{m}[\nabla f_1(x_1), \cdots, \nabla f_m(x_m)]^\top.$$

Moreover, we denote the local proximal operator and aggregate proximal operator as

$$\mathbf{prox}_{\eta,r}(x) = \operatorname*{argmin}_{z \in \mathbb{R}^d}\left(r(z) + \frac{1}{2\eta}\|z - x\|^2\right), \quad \mathbf{prox}_{m\eta,R}(\mathbf{x}) = \operatorname*{argmin}_{\mathbf{z} \in \mathbb{R}^{m \times d}}\left(R(\mathbf{z}) + \frac{1}{2m\eta}\|\mathbf{z} - \mathbf{x}\|^2\right). \tag{4}$$

# 3  Decentralized Accelerated Proximal Gradient Descent

In this section, we propose a novel decentralized proximal gradient descent algorithm achieving the optimal computational complexity and near optimal communication complexity. We describe our decentralized accelerated proximal gradient descent method (DAPG) in Algorithm 1. The use of multi-consensus (FastMix) and gradient tracking (that is, each time we track the gradient by communicating $s_t + \nabla F(\mathbf{y}_{t+1}) - \nabla F(\mathbf{y}_t)$) is motivated by Mudag of [26]. However, the actual algorithm is quite different, due to the need to handle the proximal term.

---

**Algorithm 1** DPAG

1: **Input:** $\mathbf{x}_0^{(i)} = \mathbf{x}_0^{(j)}$ for $1 \le i, j, \le m$, $\mathbf{y}_0 = \mathbf{x}_0$, $\mathbf{s}_0 = \nabla F(\mathbf{x}_0)$, $\eta = \frac{1}{L}$ and $\alpha = \sqrt{\frac{\mu}{L}}$, $K = \mathcal{O}\left(\frac{1}{\sqrt{1-\lambda_2(W)}} \log \frac{p(M,L,\mu)}{q(M,L,\mu)}\right)$, where $p, q$ are polynomials with order less than 3.
2: **for** $t = 0, \ldots, T$ **do**
3: $\quad \mathbf{x}_{t+1} = \text{FastMix}(\mathbf{prox}_{\eta m, R}(\mathbf{y}_t - \eta \mathbf{s}_t), K)$
4: $\quad \mathbf{y}_{t+1} = \text{FastMix}\left(\mathbf{x}_{t+1} + \frac{1-\alpha}{1+\alpha}(\mathbf{x}_{t+1} - \mathbf{x}_t), K\right)$
5: $\quad \mathbf{s}_{t+1} = \text{FastMix}(\mathbf{s}_t + \nabla F(\mathbf{y}_{t+1}) - \nabla F(\mathbf{y}_t), \ K)$
6: **end for**
7: **Output:** $\bar{x}_T$.

---

**Algorithm 2** FastMix

1: **Input:** $\mathbf{x}^0 = \mathbf{x}^{-1}$, $K$, $W$, step size $\eta_w = \frac{1-\sqrt{1-\lambda_2^2(W)}}{1+\sqrt{1-\lambda_2^2(W)}}$.
2: **for** $k = 0, \ldots, K$ **do**
3: $\quad \mathbf{x}^{k+1} = (1 + \eta_w)W\mathbf{x}^k - \eta_w \mathbf{x}^{k-1}$;
4: **end for**
5: **Output:** $\mathbf{x}^K$.

---

### 3.1 Sketch of the Main Proof Techniques

The main idea behind DAPG is trying to approximate the centralized accelerated proximal gradient descent by Nesterov's acceleration, multi-consensus, and gradient tracking. We first define the average of the aggregated variables defined in Algorithm 1 as follows

$$\bar{x}_t = \frac{1}{m}\sum_{i=0}^{m}\mathbf{x}_t^{(i)}, \quad \bar{y}_t = \frac{1}{m}\sum_{i=0}^{m}\mathbf{y}_t^{(i)}, \quad \bar{s}_t = \frac{1}{m}\sum_{i=0}^{m}\mathbf{s}_t^{(i)}, \quad \bar{g}_t = \frac{1}{m}\sum_{i=0}^{m}\nabla f_i(\mathbf{y}_t^{(i)}), \quad (5)$$

where $\mathbf{x}^{(i)}, \mathbf{y}^{(i)}$ indicates the $i$-th row of matrix $\mathbf{x}$ and $\mathbf{y}$. We can regard these averaged variables as the approximation of their centralized counterparts.

In our method, instead of solving Problem (1) over the network, we minimize the reformulated function (3). Our algorithm tries to use accelerated proximal gradient descent to minimize $H(\mathbf{x})$ where the acceleration helps to achieve a fast convergence rate. To deal with the consensus constraints $x_i = x_j$ for $1 \le i, j \le m$, we resort to the multi-consensus and gradient-tracking techniques. We then show that variables such as $\mathbf{x}_t^i, \mathbf{y}_t^i$ and $\mathbf{s}_t^i$ in agent $i$ will converge to their centralized counterparts $\bar{x}_t, \bar{y}_t$ and $\bar{s}_t$ as $t$ increases.

Once local variables and gradients can approximate their centralized counterparts, we can show that our algorithm has convergence properties similar to that of the centralized accelerated proximal gradient descent. This implies that our algorithm can also achieve the optimal computational complexity [13].

Previously multi-consensus was regarded as communication-unfriendly because, without gradient-tracking, one has to increase the communication times to achieve high precision consensuses [6, 14]. However, it was shown in [26] that the combination of multi-consensus and gradient-tracking leads to communication-efficiency. We follow this argument, and show that the proposed algorithm also achieves near optimal communication complexity in the proximal case.

### 3.2 Complexity Analysis

Following earlier work, we measure the computational complexity by the number of times that the gradient of $f(x)$ is computed, and we measure the communication complexity by the times of local communications, which is presented as $Wx$ in our algorithm. Similar to the analysis of accelerated proximal gradient descent method, we define the Lyapunov function as follows

$$V_t \triangleq h(\bar{x}_t) - h(x^*) + \frac{\mu}{2}\|\bar{v}_t - x^*\|^2, \quad (6)$$

where $\bar{v}_t = \bar{x}_{t-1} + \frac{1}{\alpha}(\bar{x}_t - \bar{x}_{t-1})$, with $\alpha = \sqrt{\frac{\mu}{L}}$.

In the following lemma, we will show how the multi-consensus ('FastMix' operation in Algorithm 2) helps to achieve the consensus and bound the error between local variables and their centralized counterparts.

**Lemma 1.** *Suppose* $f(x)$ *is* $L$-smooth *and* $\mu$-strongly convex. *Assume each* $f_i(x)$ *is* $M$-smooth *and* $r$ *is a proper and lower-semicontinuous convex function. Let* $\mathbf{z}_t = [\|\mathbf{x}_t - \mathbf{1}\bar{x}_t\|, \|\mathbf{y}_t - \mathbf{1}\bar{y}_t\|, \|\mathbf{s}_t - \mathbf{1}\bar{s}_t\|]^\top$, *where* $\mathbf{1} = [1, \ldots, 1]^\top \in \mathbb{R}^m$. *It holds that*

$$\mathbf{z}_{t+1} = \mathbf{A}\mathbf{z}_t + 8\rho M \sqrt{m}[0, 0, \sqrt{\frac{2}{\mu}V_t}]^\top, \tag{7}$$

*where* $\rho$ *and* $\mathbf{A}$ *are defined as*

$$\rho = \left(1 - \sqrt{1 - \lambda_2(W)}\right)^K, \quad \mathbf{A} \triangleq \rho \cdot \begin{pmatrix} 0 & 2 & 2\eta \\ 2 & 4 & 4\eta \\ 2M & M(13 + 4M\eta) & 1 + 6M\eta \end{pmatrix}$$

*Furthermore, we have*

$$\mathbf{z}_{t+1} \leq \mathbf{A}^{t+1}\mathbf{z}_0 + 8M\rho\sqrt{\frac{2m}{\mu}} \cdot \sum_{i=0}^{t} \mathbf{A}^{t-i}[0, 0, \sqrt{V_i}]^\top.$$

If the spectral norm of $\mathbf{A}$ is less than 1 then $V_t$ will converge to zero. It follows that $\|\mathbf{z}_t\|$, which can be regarded as a measure of total decentralized error compared to its centralized counterpart, will converge to zero. That is, the proposed method can well approximate accelerated proximal gradient descent. The following Lemmas show how the Lyapunov function $V_t$ decreases at the disturbance caused by the decentralized setting.

**Lemma 2.** *The Lyapunov function* $V_t$ *of Algorithm 1 defined in Eqn.* (6) *has the following property*

$$V_{t+1} \leq (1 - \alpha)V_t + D_1\sqrt{V_t} \cdot \|\mathbf{z}_t\| + D_2 \|\mathbf{z}_t\|^2, \tag{8}$$

*where* $\mathbf{z}_t$ *is defined in Lemma 1 and the constants* $D_1$, $D_2$ *are defined as follows*

$$D_1 = \frac{4}{\sqrt{m}}\left(21L + 2\sqrt{L\mu}(2 + 2M\eta) + 3\mu(2 + 2M\eta) + 8 + \frac{\mu}{L} + 3\sqrt{\frac{\mu}{L}}\right)$$

$$D_2 = \frac{8}{m}(1 + \mu) \cdot \left(9 \cdot (12 + 8L + 4M)^2 + 16L(L + 1) + \frac{133 + 79L}{L}\right).$$

By Lemma 1, we can obtain that the convergence properties of $\|\mathbf{z}_t\|$ is determined by the value of $\rho$ and $V_t$. At the same time, Lemma 2 shows that the value of $\|\mathbf{z}_t\|$ will affect the convergence properties of $V_t$ in turn. In the following lemma, we give the condition to achieve $\|\mathbf{A}\|_2 \leq \frac{1}{2}$.

**Lemma 3.** *Let* $\|\mathbf{A}\|_2$ *be the spectral norm of matrix* $\mathbf{A}$ *defined in Lemma 1. With the constant* $D_3$ *defined as*

$$D_3 = 9 + 6\eta + 15M + 4M^2\eta + 6M\eta,$$

*if* $\rho < \frac{1}{2D_3}$, *we will have* $\|\mathbf{A}\|_2 < \frac{1}{2}$.

In the following lemma, we will show that once $\alpha \leq \frac{1}{2}$, the Lyapunov function $V_t$ will converge with the rate $1 - \frac{\alpha}{2}$ once we choose a proper $\rho$. We can observe that we can get a slightly slower convergence rate than vanilla accelerated proximal gradient descent and this is caused by the error brought by the decentralized setting.

**Lemma 4.** *Assuming that* $\alpha \leq \frac{1}{2}$, $f(x)$ *is* $L$-smooth *and* $\mu$-strongly convex, *each* $f_i(x)$ *is* $M$-smooth, $r(x)$ *is a proper and lower-semicontinuous convex function, then Algorithm 1 has the following convergence rate*

$$V_{t+1} \leq \left(1 - \frac{\alpha}{2}\right)^{t+1}(V_0 + C\|\mathbf{z}_0\|^2), \text{ with } C = \frac{D_1^2}{\alpha} + 2D_2, \tag{9}$$

*if* $\rho$ *satisfies the conditions in Lemma 3 and*

$$\rho < \frac{\alpha}{2(D_1D_4 + D_2D_4^2)}, \text{ with } D_4 = 24M\sqrt{\frac{2m}{\mu}} + 2\frac{D_3}{\sqrt{C}}.$$

*Moreover, we can bound* $\|\mathbf{z}_t\|$ *as*

$$\|\mathbf{z}_t\|^2 \leq 2\rho \cdot D_4^2\left(1 - \frac{\alpha}{2}\right)^t\left(V_0 + C\|\mathbf{z}_0\|^2\right). \tag{10}$$

With the above core lemmas, we give the detailed computation complexity and communication complexity of our algorithm in the following theorem.

**Theorem 1** (Main Theorem). *Assume that $f(x)$ is $L$-smooth and $\mu$-strongly convex, each $f_i(x)$ is $M$-smooth, $\alpha \leq \frac{1}{2}$, $r$ is a proper and lower-semicontinuous convex function. Letting $K$ satisfy that*

$$K = \sqrt{\frac{\kappa_g}{1 - \lambda_2(W)}} \log(\rho^{-1}), \text{ with } \rho < \min\left\{\frac{\alpha}{2(D_1 D_4 + D_2 D_4^2)}, \frac{1}{2D_3}\right\},$$

*then, it holds that for the output $\bar{x}_T$ of Algorithm 1,*

$$h(\bar{x}_T) - h(x^*) \leq \left(1 - \frac{\alpha}{2}\right)^T \left(h(\bar{x}_0) - h(x^*) + \frac{\mu\|\bar{x}_0 - x^*\|^2}{2} + C\sum_{i=1}^{m}\|\nabla f_i(\bar{x}_0) - \nabla f(\bar{x}_0)\|^2\right).$$

*To achieve $h(\bar{x}_T) - h(x^*) < \epsilon$ and $\|\mathbf{x}_T - \mathbf{1}x^*\|^2 = \mathcal{O}(\epsilon/\mu)$, the computational and communication complexities of Algorithm 1 are*

$$T = \mathcal{O}\left(\sqrt{\kappa_g}\log\frac{1}{\epsilon}\right), \text{ and } Q = \mathcal{O}\left(\sqrt{\frac{\kappa_g}{1 - \lambda_2(W)}}\log\frac{p(M, L, \mu)}{q(M, L, \mu)}\log(\frac{1}{\epsilon})\right),$$

*where each $\mathcal{O}(\cdot)$ contains a universal constant and $p(M, L, \mu), q(M, L, \mu)$ are polynomials with order less than 3.*

*Proof.* Because $\rho$ satisfies the conditions in Lemma 4, combining with the definition of $V_t$ in Eqn. (6), we obtain that

$$h(\bar{x}_T) - h(x^*) \leq V_T \leq \left(1 - \frac{1}{2}\sqrt{\frac{\mu}{L}}\right)^T \left(h(\bar{x}_0) - h(x^*) + \frac{\mu}{2}\|\bar{x}_0 - x^*\|^2 + C\|\mathbf{z}_0\|^2\right)$$

$$\leq \exp\left(-\frac{T}{2}\sqrt{\frac{\mu}{L}}\right)\left(h(\bar{x}_0) - h(x^*) + \frac{\mu}{2}\|\bar{x}_0 - x^*\|^2 + C\|\mathbf{z}_0\|^2\right),$$

where $C$ is defined in Lemma 4. Because when $t = 0$ we set all nodes as the same status, i.e., $\mathbf{x}_0 = \mathbf{1}\bar{x}_0$ and $\mathbf{y}_0 = \mathbf{1}\bar{y}_0$, we have $\|\mathbf{z}_0\|^2 = \sum_{i=1}^m \|\nabla f_i(\bar{x}_0) - \nabla f_i(x^*)\|^2$. Thus, to achieve $h(\bar{x}_T) - h(x^*) < \epsilon$, $T$ requires to be

$$T = 2\sqrt{\kappa_g}\log\frac{h(\bar{x}_0) - h(x^*) + \frac{\mu}{2}\|\bar{x}_0 - x^*\|^2 + C\sum_{i=1}^m\|\nabla f_i(\bar{x}_0) - \nabla f_i(x^*)\|^2}{\epsilon} = \mathcal{O}(\sqrt{\kappa_g}\log\frac{1}{\epsilon}).$$

By Eqn. (10) of Lemma 4, we have

$$\|\mathbf{z}_T\|^2 \leq 2\rho^2 D_4^2 \left(1 - \frac{\alpha}{2}\right)^T (V_0 + C\|\mathbf{z}_0\|)^2 = \mathcal{O}(\epsilon/\mu).$$

Therefore, we can obtain

$$\|\mathbf{x}_T - \mathbf{1}x^*\|^2 \leq 2(\|\mathbf{x}_T - \mathbf{1}\bar{x}_T\|^2 + \|\mathbf{1}\bar{x}_T - \mathbf{1}x^*\|^2) \leq \|\mathbf{z}_T\|^2 + \frac{4}{\mu}V_T = \mathcal{O}(\epsilon/\mu).$$

The bound of $K$ can be obtained by Proposition 1. By the properties of $D_1, D_2, D_3, C, D_4$, it is easy to check that $D_3$ and $\frac{\alpha}{2(D_1 D_4 + D_2 D_4^2)}$ are independent of $m$. Thus, to achieve the conditions of $\rho$ in Lemma 2, 3, and 4, $\rho$ only needs to be less than $\frac{p}{q}$, where $p(M, L, \mu)$ and $q(M, L, \mu)$ are polynomials of $M$, $L$ and $\mu$ with orders less than 3. Thus, by Proposition 1, we only require that $K \leq \frac{1}{\sqrt{1 - \lambda_2(W)}}\log\frac{p}{q}$. Combining with the computation complexity, we can obtain the total communication complexity as

$$Q = \mathcal{O}\left(\sqrt{\frac{\kappa_g}{1 - \lambda_2(W)}}\log\frac{p(M, L, \mu)}{q(M, L, \mu)}\log\frac{1}{\epsilon}\right).$$

$\square$

**Remark 1.** *We can observe that DAPG establishes the optimal computational complexity [13] and a near optimal communication complexity which matches the lower bound up to a $\log$ term independent of $\epsilon$ [16]. This is the best computation and communication complexity of decentralized proximal algorithms can achieve. Before our work, the fastest decentralized proximal algorithm is NIDS which establishes computation and communication complexities both of $\mathcal{O}\left(\max\left\{\kappa_\ell, \frac{1}{1 - \lambda_2(W)}\right\}\right)$ [9, 25]. We can observe that the complexities of DAPG are much less than that of NIDS.*

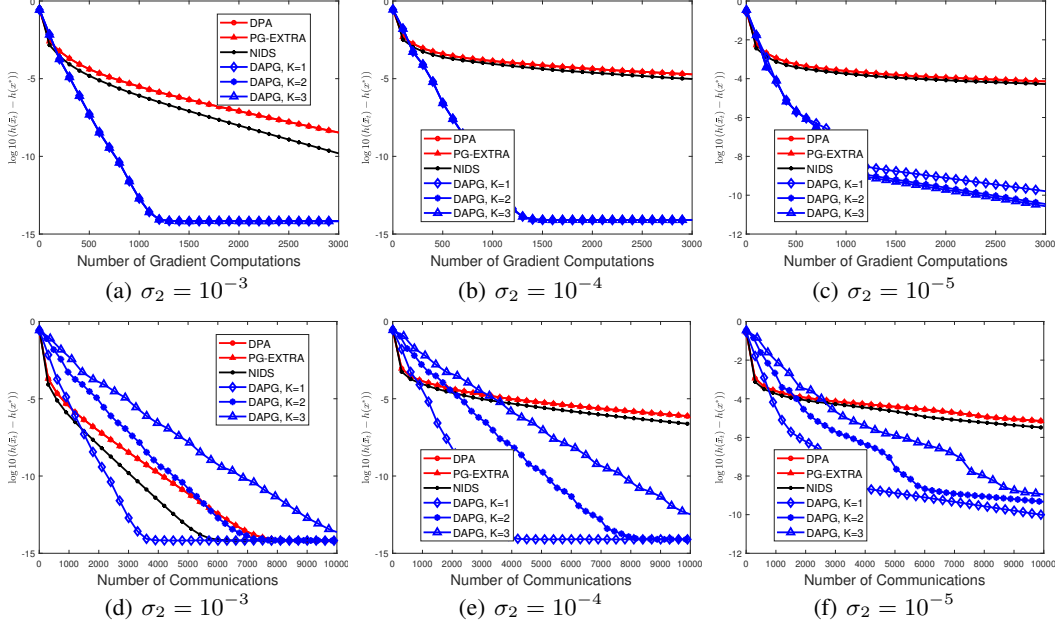

Figure 1: Experiment on 'a9a'. We compare the computation cost of algorithms in the top row and compare the communication cost of algorithms in the bottom row.

**Remark 2.** *The complexities of* DAPG *are linear to* $\sqrt{\kappa_g}$ *instead of* $\kappa_\ell$. *In contrast, the complexities of algorithms fitting into the framework of [25] are linear to* $\kappa_\ell$. *Note that* $\kappa_\ell$ *can be infinitely larger than* $\kappa_g$. *For example, for* $f(x) = \frac{1}{2}(f_1(x) + f_2(x)), x \in \mathbb{R}^2$, *with* $f_1(x) = x_1^2$ *and* $f_2(x) = x_1^2 + x_2^2$, *then it holds that* $\kappa_g = 2$ *and* $\kappa_\ell = \infty$. *Thus, the complexities of* DAPG *are better than those depending on the local condition number.*

**Remark 3.** DAPG *does not require each* $f_i(x)$ *be convex to achieve fast convergence rate. Thus,* DAPG *has a wide application range since* $f_i(x)$ *may be non-convex in some machine learning applications.* SONATA *can also be applied in these applications [21]. However,* SONATA *has to construct an SCA surrogate function of the non-convex function* $f_i(x)$ *to deal with the local non-convexity. Furthermore,* DAPG *only takes a cheap proximal mapping each iteration while* SONATA *has to minimize a sub-problem. Hence,* DAPG *is a simpler and easier to implement algorithm.*

## 4 Experiments

In the previous sections, we have given the theoretical analysis of our algorithm. In this section, we will validate the effectiveness and computational efficiency of our algorithm empirically. We will conduct experiments on the sparse logistic regression problem where $f(x)$ is general strongly convex. The sparse logistic regression is defined as

$$F(x) = \frac{1}{m}\sum_{i=1}^{m} f_i(x) + \sigma_1\|x\|_1 + \frac{\sigma_2}{2}\|x\|^2, \text{ with } f_i(x) = \frac{1}{n}\sum_{j=1}^{n}\log[1 + \exp(-b_j\langle a_j, x\rangle)] \quad (11)$$

where $a_j \in \mathbb{R}^d$ is the $j$-th input vector, and $b_j \in \{-1, 1\}$ is the corresponding label.

**Experiments Setting** In our experiments, we consider random networks where each pair of agents have a connection with a probability of $p = 0.1$. We set $W = I - \frac{L}{\lambda_1(L)}$ where $L$ is the Laplacian matrix associated with a weighted graph, and $\lambda_1(L)$ is the largest eigenvalue of $L$. We set $m = 100$, that is, there exists 100 agents in this network. In our experiments, the gossip matrix $W$ satisfies $1 - \lambda_2(W) = 0.05$.

We conduct experiments on the datasets 'w8a' and 'w9a' which can be downloaded in libsvm datasets. For 'w8a', we set $n = 497$ and $d = 300$. For 'a9a', we set $n = 325$ and $d = 123$. We set $\sigma_1 = 10^{-4}$

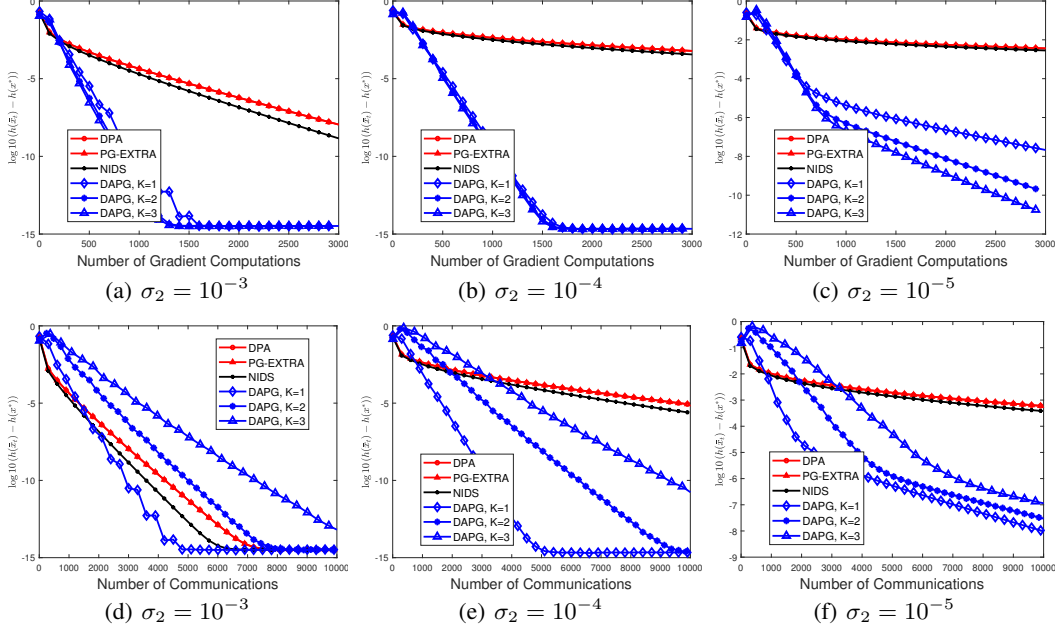

Figure 2: Experiment on 'w8a'. We compare the computation cost of algorithms in the top row and compare the communication cost of algorithms in the bottom row.

for all datasets and set $\sigma_2$ as $10^{-3}$, $10^{-4}$ and $10^{-5}$ to control the condition number of the objective function.

**Comparison with Existing Works**  We compare our work with state-of-the-art algorithms PG-EXTRA [19], NIDS [9] and Decentralized Proximal Algorithm (DPA) [2]. In the experiments, we set $K = 1, 2, 3$ respectively. The parameters of all algorithms are well-tuned. We report experiment results in Figure 1 and 2. We can observe that DAPG takes much less computational cost than other algorithms because DAPG uses Nesterov's acceleration to achieve a faster convergence rate. This matches our theoretical analysis of the computation complexity. We can further observe that the advantage of DAPG is more clear when $\sigma_2$ is small. This is because a small $\sigma_2$ commonly leads to a large condition number and the computation complexity of DAPG is linear to $\sqrt{\kappa_g}$ instead of $\kappa_\ell$. DAPG also shows great advantages over other state-of-the-art decentralized proximal algorithms on the communication cost. Though DAPG takes three times of local communication while other algorithms communicate only once for each iteration, DAPG still requires much less communication costs because of its fast convergence rate when $\sigma_2$ is small.

# 5   Conclusion

In this paper, we studied the decentralized composite optimization problem with a non-smooth regularization term. We proposed a novel algorithm that achieves the optimal computation complexity and a near optimal communication complexity matching the lower bound up to a $\log$ term independent of $\epsilon$. This is the best known communication complexity that decentralized proximal gradient algorithms can achieve. Furthermore, DAPG does not require each individual function $f_i(x)$ to be convex to achieve a fast convergence rate. Hence, our algorithm has a wider range of applications in machine learning than other decentralized proximal gradient descent algorithms which require $f_i(x)$ to be convex. Finally, the complexities of our algorithms depend on the global condition number $\kappa_g$ instead of the local one. Since $\kappa_g \leq \kappa_\ell$ and $\kappa_\ell$ can be infinitely larger than $\kappa_g$, our algorithm can achieve much better performance than the algorithms whose complexities depend on $\kappa_\ell$. The experiments also validate the advantages of our algorithm.

# 6 Broader Impact

Our work focuses on the theory of decentralized optimization and proposes a novel decentralized proximal algorithm. This will help us to design new decentralized proximal algorithms. Because decentralized optimization has wide applications in machine learning, sensor networks, and multi-robot system, our work may be used in these areas.

## Acknowledgments and Disclosure of Funding

This work is supported by the project of Shenzhen Research Institute of Big Data (named "Automated Machine Learning") and GRF 16201320.

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
