[Supplementary Material]

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

# A  Proof of Proposition 1

*Proof of Proposition 1.* By the update rule of Algorithm 2 and the fact that $W^\infty = \frac{1}{m}\mathbf{1}\mathbf{1}^\top$ [23], we have

$$W^\infty \mathbf{x}^K = W^\infty \mathbf{x}^{K-1} + \eta_w \left( W^\infty \mathbf{x}^{K-1} - W^\infty \mathbf{x}^{K-2} \right).$$

We can obtain that

$$W^\infty \left( \mathbf{x}^K - \mathbf{x}^{K-1} \right) = \eta_w \left( W^\infty \mathbf{x}^{K-1} - W^\infty \mathbf{x}^{K-2} \right).$$

Note that $\mathbf{x}^0 = \mathbf{x}^{-1}$ in Algorithm 2, we can obtain that for any $k = 0, \dots, K$, we have

$$W^\infty \left( \mathbf{x}^k - \mathbf{x}^{k-1} \right) = 0.$$

Therefore, we can obtain the identity $W^\infty \mathbf{x}^K = W^\infty \mathbf{x}^0$, which implies the result. The convergence rate of Algorithm 2 can be found in [10]. $\qquad\square$

# B  Collection of Lemmas

We list several important lemmas that will be used in our proofs.

**Lemma 5** ([13]). *Letting $\tilde{\nabla}h(x)$ the generalized gradient of $h(x)$ defined as*

$$\tilde{\nabla}h(x) \triangleq \frac{x - \boldsymbol{prox}_{\eta,r}(x - \eta \nabla f(x))}{\eta}, \quad \text{with } \eta = \frac{1}{L} \text{ being the step size,} \qquad (12)$$

*then it holds that $\tilde{\nabla}h(x^*) = 0$ if $x^*$ minimizes $h(x)$.*

**Lemma 6.** *Letting $\boldsymbol{prox}^{(i)}_{\eta m,R}(\mathbf{x})$ denote the $i$-th row of the matrix $\boldsymbol{prox}_{\eta m,R}(\mathbf{x})$ (defined in Eqn. (4)), we have the following equation*

$$\boldsymbol{prox}^{(i)}_{\eta m,R}(\mathbf{x}) = \boldsymbol{prox}_{\eta,r}(\mathbf{x}^{(i)}).$$

*Proof.* By the definition of the proximal operators, we have

$$\begin{aligned}
\mathbf{prox}_{\eta m,R}(\mathbf{x}) &= \underset{\mathbf{z}}{\arg\min} \left( R(\mathbf{z}) + \frac{1}{2\eta m} \|\mathbf{z} - \mathbf{x}\|_F^2 \right) \\
&= \underset{\mathbf{z}}{\arg\min} \left( \frac{1}{m} \sum_{i=1}^{m} r(\mathbf{z}^{(i)}) + \sum_{i=1}^{m} \frac{1}{2\eta m} \left\| \mathbf{z}^{(i)} - \mathbf{x}^{(i)} \right\|^2 \right) \\
&= \underset{\mathbf{z}}{\arg\min} \left( \sum_{i=1}^{m} r(\mathbf{z}^{(i)}) + \sum_{i=1}^{m} \frac{1}{2\eta} \left\| \mathbf{z}^{(i)} - \mathbf{x}^{(i)} \right\|^2 \right) \\
&= \begin{pmatrix} \arg\min_z \left( r(z) + \frac{1}{2\eta} \left\| z - \mathbf{x}^{(1)} \right\| \right)^\top \\ \vdots \\ \arg\min_z \left( r(z) + \frac{1}{2\eta} \left\| z - \mathbf{x}^{(m)} \right\| \right)^\top \end{pmatrix}.
\end{aligned}$$

Therefore, we have the following equation

$$\mathbf{prox}^{(i)}_{\eta m,R}(\mathbf{x}) = \mathbf{prox}_{\eta,r}(\mathbf{x}^{(i)}).$$

$\qquad\square$

**Lemma 7.** *For $\bar{x}_t$, $\bar{y}_t$ and $\bar{v}_t$ defined in Eqn. (5) and (6), then we can obtain that*

$$\bar{y}_t - \bar{x}_t = \alpha(\bar{v}_t - \bar{y}_t)$$
$$\bar{y}_{t+1} = \frac{\bar{x}_{t+1} + \alpha \bar{v}_{t+1}}{1 + \alpha}.$$

*Proof.* First using the definition of $\bar{v}_t$ we have

$$\frac{\bar{x}_{t+1} + \alpha\bar{v}_{t+1}}{1 + \alpha} = \frac{\bar{x}_{t+1} + \alpha[\bar{x}_t + \frac{1}{\alpha}(\bar{x}_{t+1} - \bar{x}_t)]}{1 + \alpha}$$

$$= \bar{x}_{t+1} + \frac{1 - \alpha}{1 + \alpha}(\bar{x}_{t+1} - \bar{x}_t)$$

$$= \bar{y}_{t+1}$$

Then we can have

$$\bar{y}_t - \bar{x}_t = \alpha(\bar{v}_t - \bar{y}_t).$$

$\square$

**Lemma 8.** *Let $h(x)$ be $\mu$-strongly convex. For $\bar{y}_t$, $x^*$ and $V_t$ defined in Eqn. (6) and (5), we have the following inequality,*

$$\|\bar{y}_t - x^*\| \leq \sqrt{\frac{2}{\mu}V_t}.$$

*Proof.* Combined Lemma 7, the optimality of $x^*$, and the $\mu$-strongly convexity of $h(x)$, we obtain that

$$\|\bar{y}_t - x^*\| = \left\|\frac{\bar{x}_t + \alpha\bar{v}_t}{1 + \alpha} - x^*\right\| \leq \frac{1}{1 + \alpha}\|\bar{x}_t - x^*\| + \frac{\alpha}{1 + \alpha}\|\bar{v}_t - x^*\| \leq \sqrt{\frac{2}{\mu}V_t}.$$

$\square$

**Lemma 9.** *If $r$ is a proper and semi-continuous function, for any $z$, $w$ and $v = \mathbf{prox}_{\eta,r}(w)$, it holds that*

$$r(v) \leq r(z) + \frac{1}{\eta}(v - w)^\top(z - v).$$

*Proof.* For any given $w$, we have

$$v = \mathbf{prox}_{\eta,r}(w) = \underset{v}{\mathrm{argmin}} \frac{1}{2\eta}\|v - w\|^2 + r(v).$$

Thus, we can obtain that

$$0 \in \partial\left(\frac{1}{2\eta}\|v - w\|^2 + r(v)\right) = -\frac{1}{\eta}(w - v) + \partial r(v),$$

where $\partial$ here denotes the subgradient. Then we can get that

$$\frac{1}{\eta}(w - v) \in \partial r(v).$$

According to the definition of subgradient, we have for all $z$,

$$r(z) \geq r(v) - \frac{1}{\eta}(v - w)^\top(z - v).$$

Thus we obtain

$$r(v) \leq r(z) + \frac{1}{\eta}(v - w)^\top(z - v),$$

for all $z$, $w$ and $v = \mathbf{prox}_{\eta,r}(w)$.

$\square$

**Lemma 10** (Properties of gradients). *Letting $f(x)$ be $L$-smooth and each $f_i(x)$ to be $M$-smooth, we have following properties on the aggregate gradients and generalized gradients.*

$$\|\nabla F(\mathbf{y}) - \nabla F(\mathbf{x})\| \leq M\|\mathbf{y} - \mathbf{x}\|, \tag{13}$$

$$\|\bar{g}_t - \nabla f(\bar{y}_t)\| \leq \frac{M}{\sqrt{m}}\|\mathbf{y}_t - \mathbf{1}\bar{y}_t\|. \tag{14}$$

*Furthermore, if we set $\eta = \frac{1}{L}$, we have the $(3L)$-smooth property for the generalized gradient (defined in Eqn. (12))*

$$\left\|\tilde{\nabla}h(x) - \tilde{\nabla}h(y)\right\| \leq \left(\frac{2}{\eta} + L\right)\|x - y\| = 3L\|x - y\|. \tag{15}$$

*Proof.* Eqn. (13) can be obtained from the $M$-smoothness of the local functions $f_i(x)$ as follows

$$\|\nabla F(\mathbf{y}) - \nabla F(\mathbf{x})\| = \sqrt{\sum_i^m \left\|\nabla f_i(\mathbf{y}^{(i)}) - \nabla f_i(\mathbf{x}^{(i)})\right\|^2}$$

$$\leq \sqrt{M^2 \sum_i^m \left\|\mathbf{y}^{(i)} - \mathbf{x}^{(i)}\right\|^2}$$

$$= M \|\mathbf{y} - \mathbf{x}\| .$$

To prove the Eqn. (14), we have

$$\|\bar{g}_t - \nabla f(\bar{y}_t)\| = \left\| \frac{1}{m} \sum_{i=0}^m \left[ \nabla f_i(\mathbf{y}_t^{(i)}) - \nabla f_i(\bar{y}_t) \right] \right\|$$

$$= \left\| \sum_{i=1}^m \frac{\nabla f_i(\mathbf{y}_t^{(i)}) - \nabla f_i(\bar{y}_t)}{m} \right\|$$

$$\leq M \sum_{i=0}^m \frac{\left\| \mathbf{y}_t^{(i)} - \bar{y}_t \right\|}{m}$$

$$\leq M \sqrt{\sum_{i=0}^m \frac{\left\| \mathbf{y}_t^{(i)} - \bar{y}_t \right\|^2}{m}}$$

$$= M \frac{1}{\sqrt{m}} \|\mathbf{y}_t - \mathbf{1}\bar{y}_t\| ,$$

where the first inequality is due to $M$-smoothness of $f_i$, and the second inequality is because of Jensen's inequality. Then we can prove Eqn. (15) using $L$-smoothness of $f(x)$ and the non-expansiveness of proximal operator

$$\left\| \tilde{\nabla} h(x) - \tilde{\nabla} h(y) \right\| = \left\| \frac{x - \mathbf{prox}_{\eta,r}(x - \eta \nabla f(x))}{\eta} - \frac{y - \mathbf{prox}_{\eta,r}(y - \eta \nabla f(y))}{\eta} \right\|$$

$$\leq \frac{1}{\eta} \|x - y\| + \frac{1}{\eta} \left\| \mathbf{prox}_{\eta,r}(x - \eta \nabla f(x)) - \mathbf{prox}_{\eta,r}(y - \eta \nabla f(y)) \right\|$$

$$\leq \frac{1}{\eta} \|x - y\| + \frac{1}{\eta} \|(x - \eta \nabla f(x)) - (y - \eta \nabla f(y))\|$$

$$\leq \left( \frac{2}{\eta} + L \right) \|x - y\|$$

$$= 3L \|x - y\| ,$$

where the last inequality is due to the $L$-smoothness of $f(x)$.

$\square$

**Lemma 11.** *Let $\mathbf{prox}_{m\eta,R}(\cdot)$ denote the proximal operator defined in Eqn.* (4). *For any $\mathbf{x} \in \mathbb{R}^{m \times d}$, we have*

$$\left\| \mathbf{prox}_{\eta m,R}(\frac{1}{m}\mathbf{1}\mathbf{1}^\top \mathbf{x}) - \frac{1}{m}\mathbf{1}\mathbf{1}^\top \mathbf{prox}_{\eta m,R}(\mathbf{x}) \right\| \leq \|\mathbf{x} - \mathbf{1}\bar{x}_t\| .$$

*Proof.* Using Lemma 6, we expand the sum

$$\left\|\mathbf{prox}_{\eta m,R}(\frac{1}{m}\mathbf{1}\mathbf{1}^\top\mathbf{x}) - \frac{1}{m}\mathbf{1}\mathbf{1}^\top\mathbf{prox}_{\eta m,R}(\mathbf{x})\right\|$$

$$=\sqrt{\left\|\mathbf{prox}_{\eta m,R}(\frac{1}{m}\mathbf{1}\mathbf{1}^\top\mathbf{x}) - \frac{1}{m}\mathbf{1}\mathbf{1}^\top\mathbf{prox}_{\eta m,R}(\mathbf{x})\right\|^2}$$

$$=\sqrt{m\left\|\mathbf{prox}_{\eta,r}(\frac{1}{m}\mathbf{1}^\top\mathbf{x}) - \frac{1}{m}\sum_{i=1}^{m}\mathbf{prox}_{\eta,r}(\mathbf{x}^{(i)})\right\|^2}.$$

Using the inequality $(\sum_{i=1}^{m} a_i)^2 \leq m\sum_{i=1}^{m} a_i^2$, and the non-expansiveness of proximal operator, we can obtain that

$$\sqrt{m\left\|\mathbf{prox}_{\eta,r}(\frac{1}{m}\mathbf{1}^\top\mathbf{x}) - \frac{1}{m}\sum_{i=1}^{m}\mathbf{prox}_{\eta,r}(\mathbf{x}^{(i)})\right\|^2}$$

$$=\sqrt{m\left\|\frac{1}{m}\sum_{i=1}^{m}\left(\mathbf{prox}_{\eta,r}(\frac{1}{m}\mathbf{1}^\top\mathbf{x}) - \mathbf{prox}_{\eta,r}(\mathbf{x}^{(i)})\right)\right\|^2}$$

$$\leq\sqrt{\sum_{i=1}^{m}\left\|\left(\mathbf{prox}_{\eta,r}(\frac{1}{m}\mathbf{1}^\top\mathbf{x}) - \mathbf{prox}_{\eta,r}(\mathbf{x}^{(i)})\right)\right\|^2}$$

$$\leq\sqrt{\sum_{i=1}^{m}\left\|\left(\frac{1}{m}\mathbf{1}^\top\mathbf{x} - \mathbf{x}^{(i)}\right)\right\|^2}$$

$$=\|\mathbf{x} - \mathbf{1}\bar{x}_t\|.$$

$\square$

**Lemma 12.** *Define the estimated generalized gradient*

$$G_t = \eta^{-1}\left(\mathbf{y}_t - \mathbf{prox}_{\eta m,R}(\mathbf{y}_t - \eta\mathbf{s}_t)\right), and, G_t^{(i)} = \eta^{-1}\left(\mathbf{y}_t^{(i)} - \mathbf{prox}_{\eta,r}(\mathbf{y}_t^{(i)} - \eta\mathbf{s}_t^{(i)})\right). \quad (16)$$

*Letting the functions satisfy the properties in Lemma 1, we have following error bound for the estimated generalized gradient*

$$\left\|\frac{\eta}{m}\mathbf{1}^\top G_t - \eta\tilde{\nabla}h(\bar{y}_t)\right\| \leq \frac{4+2M\eta}{\sqrt{m}}\|\mathbf{y}_t - \mathbf{1}\bar{y}_t\| + \frac{2\eta}{\sqrt{m}}\|\mathbf{s}_t - \mathbf{1}\bar{s}_t\|,$$

$$\left\|\frac{\eta}{m}\mathbf{1}\cdot\mathbf{1}^\top G_t - \eta\mathbf{1}\tilde{\nabla}h(\bar{y}_t)\right\| \leq (4+2M\eta)\|\mathbf{y}_t - \mathbf{1}\bar{y}_t\| + 2\eta\|\mathbf{s}_t - \mathbf{1}\bar{s}_t\|,$$

$$\left\|\eta G_t - \eta\mathbf{1}\tilde{\nabla}h(\bar{y}_t)\right\| \leq (7+2M\eta)\|\mathbf{y}_t - \mathbf{1}\bar{y}_t\| + 4\eta\|\mathbf{s}_t - \mathbf{1}\bar{s}_t\|,$$

*where the generalized gradient $\tilde{\nabla}h(\cdot)$ is defined in Eqn. (12).*

*Proof.* Using Lemma 6, we have the $i$-th index of $G_i$ is

$$\eta^{-1}\left(\mathbf{y}_t^{(i)} - \mathbf{prox}_{\eta m,R}^{(i)}(\mathbf{y}_t - \eta\mathbf{s}_t)\right)$$

$$=\eta^{-1}\left(\mathbf{y}_t^{(i)} - \mathbf{prox}_{\eta,r}(\mathbf{y}_t^{(i)} - \eta\mathbf{s}_t^{(i)})\right) \quad (17)$$

$$=G_t^{(i)}.$$

Therefore $G_t^{(i)}$ is the $i$-th index of $G_t$.

Combining with the inequality $(\sum_{i=1}^{m} a_i)^2 \le m \sum_{i=1}^{m} a_i^2$, we can obtain

$$\left\| \frac{\eta}{m} \mathbf{1}^\top G_t - \eta \tilde{\nabla} h(\bar{y}_t) \right\| = \sqrt{\left\| \frac{1}{m} \sum_{i=1}^{m} \left( \eta G_t^{(i)} - \eta \tilde{\nabla} h(\bar{y}_t) \right) \right\|^2}$$

$$\le \sqrt{\frac{1}{m} \cdot \sum_{i=1}^{m} \left\| \left( \eta G_t^{(i)} - \eta \tilde{\nabla} h(\bar{y}_t) \right) \right\|^2}.$$

By the definition of the estimated generalized gradient $G_t$ and $G_t^{(i)}$, we have

$$\sqrt{\frac{1}{m} \cdot \sum_{i=1}^{m} \left\| \left( \eta G_t^{(i)} - \eta \tilde{\nabla} h(\bar{y}_t) \right) \right\|^2}$$

$$= \sqrt{\frac{1}{m}} \cdot \sqrt{\sum_{i=1}^{m} \left\| \left( \mathbf{y}_t^{(i)} - \mathbf{prox}_{\eta,r}(\mathbf{y}_t^{(i)} - \eta \mathbf{s}_t^{(i)}) \right) - \left( \bar{y}_t - \mathbf{prox}_{\eta,r}(\bar{y}_t - \eta \nabla f(\bar{y}_t)) \right) \right\|^2}$$

$$\le \sqrt{\frac{1}{m}} \cdot \sqrt{\sum_{i=1}^{m} \left( 2 \left\| \mathbf{y}_t^{(i)} - \bar{y}_t \right\|^2 + 2 \left\| \mathbf{prox}_{\eta,r}(\mathbf{y}_t^{(i)} - \eta \mathbf{s}_t^{(i)}) - \mathbf{prox}_{\eta,r}(\bar{y}_t - \eta \nabla f(\bar{y}_t)) \right\|^2 \right)}$$

$$\le \sqrt{\frac{1}{m}} \cdot \sqrt{\sum_{i=1}^{m} \left( 2 \left\| \mathbf{y}_t^{(i)} - \bar{y}_t \right\|^2 + 2 \left\| (\mathbf{y}_t^{(i)} - \eta \mathbf{s}_t^{(i)}) - (\bar{y}_t - \eta \nabla f(\bar{y}_t)) \right\|^2 \right)}$$

$$= \sqrt{\frac{1}{m}} \cdot \sqrt{2 \left\| \mathbf{y}_t - \mathbf{1}\bar{y}_t \right\|^2 + 2 \left\| \eta \mathbf{s}_t - \eta \mathbf{1} \nabla f(\bar{y}_t) + \mathbf{y}_t - \mathbf{1}\bar{y}_t \right\|^2}$$

$$\le 4 \left\| \mathbf{y}_t - \mathbf{1}\bar{y}_t \right\| + 2\eta \left\| \mathbf{s}_t - \mathbf{1}\bar{s}_t \right\| + 2\eta \left\| \mathbf{1}\bar{s}_t - \mathbf{1}\nabla f(\bar{y}_t) \right\|$$

$$\le \sqrt{\frac{1}{m}} \cdot \left( (4 + 2M\eta) \left\| \mathbf{y}_t - \mathbf{1}\bar{y}_t \right\| + 2\eta \left\| \bar{s}_t - \mathbf{1}\bar{s}_t \right\| \right),$$

where the second inequality is due to the non-expansiveness of proximal operator, and the last inequality is from Eqn. (14). Combining above two inequalities, we can obtain that

$$\left\| \frac{\eta}{m} \mathbf{1}^\top G_t - \eta \tilde{\nabla} h(\bar{y}_t) \right\|$$

$$\le \sqrt{\frac{1}{m}} \sqrt{\sum_{i=1}^{m} \left\| \left( \eta G_t^{(i)} - \eta \tilde{\nabla} h(\bar{y}_t) \right) \right\|^2}$$

$$\le \frac{4 + 2M\eta}{\sqrt{m}} \left\| \mathbf{y}_t - \mathbf{1}\bar{y}_t \right\| + \frac{2\eta}{\sqrt{m}} \left\| \bar{s}_t - \mathbf{1}\bar{s}_t \right\|.$$

Because $\left\| \frac{\eta}{m} \mathbf{1}\mathbf{1}^\top G_t - \eta \mathbf{1}\tilde{\nabla} h(\bar{y}_t) \right\|$ is $\sqrt{m}$ times of $\left\| \frac{\eta}{m} \mathbf{1}^\top G_t - \eta \tilde{\nabla} h(\bar{y}_t) \right\|$, we can obtain the matrix version of the inequality

$$\left\| \frac{\eta}{m} \mathbf{1}\mathbf{1}^\top G_t - \eta \mathbf{1}\tilde{\nabla} h(\bar{y}_t) \right\| = \sqrt{m} \cdot \left\| \frac{\eta}{m} \mathbf{1}^\top G_t - \eta \tilde{\nabla} h(\bar{y}_t) \right\|$$

$$\le (4 + 2M\eta) \cdot \left\| \mathbf{y}_t - \mathbf{1}\bar{y}_t \right\| + 2\eta \left\| \mathbf{s}_t - \mathbf{1}\bar{s}_t \right\|.$$

Finally, combining above results, we can obtain the gap of the approximated generalized gradients

$$\left\| \eta G_t - \eta \mathbf{1} \tilde{\nabla} h(\bar{y}_t) \right\|$$

$$\leq \left\| \eta G_t - \frac{\eta}{m} \mathbf{1} \cdot \mathbf{1}^\top G_t \right\| + \left\| \frac{\eta}{m} \mathbf{1} \cdot \mathbf{1}^\top G_t - \eta \mathbf{1} \tilde{\nabla} h(\bar{y}_t) \right\|$$

$$\leq \| \mathbf{y}_t - \mathbf{1} \bar{y}_t \| + \left\| \mathbf{prox}_{\eta m, R}(\mathbf{y}_t - \eta \mathbf{s}_t) - \frac{1}{m} \mathbf{1} \mathbf{1}^\top \mathbf{prox}_{\eta m, R}(\mathbf{y}_t - \eta \mathbf{s}_t) \right\|$$

$$\quad + (4 + 2M\eta) \| \mathbf{y}_t - \mathbf{1} \bar{y}_t \| + 2\eta \| \mathbf{s}_t - \mathbf{1} \bar{s}_t \|$$

$$\leq \| \mathbf{y}_t - \mathbf{1} \bar{y}_t \| + \left\| \mathbf{prox}_{\eta m, R}(\mathbf{y}_t - \eta \mathbf{s}_t) - \mathbf{prox}_{\eta m, R}(\mathbf{1} \bar{y}_t - \eta \mathbf{1} \bar{s}_t) \right\| + 2\eta \| \mathbf{s}_t - \mathbf{1} \bar{s}_t \|$$

$$\quad + \left\| \mathbf{prox}_{\eta m, R}(\mathbf{1} \bar{y}_t - \eta \mathbf{1} \bar{s}_t) - \frac{1}{m} \mathbf{1} \mathbf{1}^\top \mathbf{prox}_{\eta m, R}(\mathbf{y}_t - \eta \mathbf{s}_t) \right\| + (4 + 2M\eta) \| \mathbf{y}_t - \mathbf{1} \bar{y}_t \|$$

$$= (7 + 2M\eta) \| \mathbf{y}_t - \mathbf{1} \bar{y}_t \| + 4\eta \| \mathbf{s}_t - \mathbf{1} \bar{s}_t \|,$$

where the second inequality is from the definition of the $G_t$ and the last inequality is from the non-expansiveness of proximal operator and Lemma 11. □

**Lemma 13.** *Letting* $\mathbf{s}_t^{(i)}$ *be the $i$-th row of* $\mathbf{s}_t$ *defined in Algorithm 1, and functions satisfy the properties described in Lemma 1, we have*

$$\sum_{i=1}^m \left\| \mathbf{s}_t^{(i)} - \nabla f(\mathbf{x}_t^{(i)}) \right\|^2 \leq 2 \| \mathbf{s}_t - \mathbf{1} \bar{s}_t \|^2 + 8M^2 \| \mathbf{1} \bar{x}_t - \mathbf{x}_t \|^2.$$

*Proof.* Using the inequality that $(a + b)^2 \leq 2a^2 + 2b^2$, we have

$$\sum_{i=1}^m \left\| \mathbf{s}_t^{(i)} - \nabla f(\mathbf{x}_t^{(i)}) \right\|^2 \leq 2 \sum_{i=1}^m \left\| \mathbf{s}_t^{(i)} - \bar{s}_t \right\|^2 + 2 \sum_{i=1}^m \left\| \bar{s}_t - \nabla f(\mathbf{x}_t^{(i)}) \right\|^2$$

$$\leq 2 \sum_{i=1}^m \left\| \mathbf{s}_t^{(i)} - \bar{s}_t \right\|^2 + 4 \sum_{i=1}^m \| \bar{s}_t - \nabla f(\bar{x}_t) \|^2 + 4 \sum_{i=1}^m \left\| \nabla f(\bar{x}_t) - \nabla f(\mathbf{x}_t^{(i)}) \right\|^2$$

$$\leq 2 \| \mathbf{s}_t - \mathbf{1} \bar{s}_t \|^2 + 4M^2 \| \mathbf{1} \bar{x}_t - \mathbf{x}_t \|^2 + 4L^2 \| \mathbf{1} \bar{x}_t - \mathbf{x}_t \|^2$$

$$\leq 2 \| \mathbf{s}_t - \mathbf{1} \bar{s}_t \|^2 + 8M^2 \| \mathbf{1} \bar{x}_t - \mathbf{x}_t \|^2,$$

where the third inequality is from Eqn. (14) and the $L$-smoothness of $f(x)$, the last inequality is due to $L \leq M$. □

## C  Proof of core lemmas

### C.1  Proof of Lemma 1

*Proof.* Reformulate the Eqn. (5) as $\bar{x}_t = \frac{1}{m} \mathbf{1}^\top \cdot \mathbf{x}_t$, $\bar{y}_t = \frac{1}{m} \mathbf{1}^\top \cdot \mathbf{y}_t$, and $\bar{s}_t = \frac{1}{m} \mathbf{1}^\top \cdot \mathbf{s}_t$, $\bar{g}_t = \frac{1}{m} \sum_{i=1}^m f_i(\mathbf{y}_t^{(i)})$, one can check the following equations by induction and similar results can be found in [26, 14]

$$\bar{y}_{t+1} = \bar{x}_{t+1} + \frac{1 - \alpha}{1 + \alpha} (\bar{x}_{t+1} - \bar{x}_t) \tag{18}$$

$$\bar{s}_{t+1} = \bar{g}_{t+1}.$$

For simplicity, we denote FastMix$(\cdot, K)$ operation as $\mathbb{T}(\cdot)$. From Proposition 1 we can know that

$$\left\| \mathbb{T}(\mathbf{x}) - \frac{1}{m} \mathbf{1} \mathbf{1}^\top \mathbf{x} \right\| \leq \rho \left\| \mathbf{x} - \frac{1}{m} \mathbf{1} \mathbf{1}^\top \mathbf{x} \right\|. \tag{19}$$

First, we have

$$
\begin{aligned}
\|\mathbf{1}\bar{x}_{t+1} - \mathbf{x}_{t+1}\| \leq & \rho \left\| \mathbf{prox}_{\eta m, R}(\mathbf{y}_t - \eta \mathbf{s}_t) - \frac{1}{m}\mathbf{1}\mathbf{1}^\top \mathbf{prox}_{\eta m, R}(\mathbf{y}_t - \eta \mathbf{s}_t) \right\| \\
\leq & \rho \left\| \mathbf{prox}_{\eta m, R}(\mathbf{y}_t - \eta \mathbf{s}_t) - \mathbf{prox}_{\eta m, R}\left(\mathbf{1}(\bar{y}_t - \eta \bar{s}_t)\right) \right\| \\
& + \rho \left\| \mathbf{prox}_{\eta m, R}\left(\mathbf{1}(\bar{y}_t - \eta \bar{s}_t)\right) - \frac{1}{m}\mathbf{1}\mathbf{1}^\top \mathbf{prox}_{\eta m, R}(\mathbf{y}_t - \eta \mathbf{s}_t) \right\| \\
\leq & \rho \|\mathbf{y}_t - \mathbf{1}\bar{y}_t\| + \rho\eta \|\mathbf{s}_t - \mathbf{1}\bar{s}_t\| + \rho \|(\mathbf{y}_t - \eta \mathbf{s}_t) - \mathbf{1}(\bar{y}_t - \eta \bar{s}_t)\| \\
\leq & 2\rho \|\mathbf{y}_t - \mathbf{1}\bar{y}_t\| + 2\rho\eta \|\mathbf{s}_t - \mathbf{1}\bar{s}_t\|,
\end{aligned}
\tag{20}
$$

where the first inequality is due to Eqn. (19), and the third inequality is because of Lemma 11 and the non-expansiveness of proximal operator.

Using the definition of $\mathbf{y}_{t+1}$ in Algorithm 1 and the property of FastMix operation, we have

$$
\begin{aligned}
\|\mathbf{y}_{t+1} - \mathbf{1}\bar{y}_{t+1}\| \leq & \frac{2\rho}{1+\alpha} \|\mathbf{x}_{t+1} - \mathbf{1}\bar{x}_{t+1}\| + \rho\frac{1-\alpha}{1+\alpha} \|\mathbf{x}_t - \mathbf{1}\bar{x}_t\| \\
\leq & 4\rho^2 \|\mathbf{y}_t - \mathbf{1}\bar{y}_t\| + 4\rho^2\eta \|\mathbf{s}_t - \mathbf{1}\bar{s}_t\| + 2\rho \|\mathbf{x}_t - \mathbf{1}\bar{x}_t\|,
\end{aligned}
$$

where the last inequality is from Eqn. (20).

Now we are going to bound the value of $\|\mathbf{s}_{t+1} - \mathbf{1}\bar{s}_{t+1}\|$. Combining the results we have obtain on $\|\mathbf{y}_{t+1} - \mathbf{1}\bar{y}_{t+1}\|$, we have

$$
\begin{aligned}
& \|\mathbf{s}_{t+1} - \mathbf{1}\bar{s}_{t+1}\| \\
\leq & \rho \|\mathbf{s}_t + \nabla F(\mathbf{y}_{t+1}) - \nabla F(\mathbf{y}_t) - \mathbf{1}\cdot(\bar{s}_t + \bar{g}_{t+1} - \bar{g}_t)\| \\
\leq & \rho \|\mathbf{s}_t - \mathbf{1}\bar{s}_t\| + \rho M \|\mathbf{y}_{t+1} - \mathbf{y}_t\| \\
\leq & \rho \|\mathbf{s}_t - \mathbf{1}\bar{s}_t\| + \rho M \|\mathbf{y}_{t+1} - \mathbf{1}\bar{y}_{t+1}\| + \rho M \|\mathbf{1}\bar{y}_{t+1} - \mathbf{1}\bar{y}_t\| + \rho M \|\mathbf{1}\bar{y}_t - \mathbf{y}_t\| \\
\leq & \rho \|\mathbf{s}_t - \mathbf{1}\bar{s}_t\| + \rho M \left(4\rho^2 \|\mathbf{y}_t - \mathbf{1}\bar{y}_t\| + 4\rho^2\eta \|\mathbf{s}_t - \mathbf{1}\bar{s}_t\| + 2\rho \|\mathbf{x}_t - \mathbf{1}\bar{x}_t\|\right) \\
& + \rho M \|\mathbf{1}\bar{y}_{t+1} - \mathbf{1}\bar{y}_t\| + \rho M \|\mathbf{1}\bar{y}_t - \mathbf{y}_t\| \\
= & \left(\rho + 4\rho^3 M\eta\right) \|\mathbf{s}_t - \mathbf{1}\bar{s}_t\| + 2\rho^2 M \|\mathbf{x}_t - \mathbf{1}\bar{x}_t\| + \left(\rho M + 4\rho^3 M\right) \|\mathbf{1}\bar{y}_t - \mathbf{y}_t\| + \rho M \|\mathbf{1}\bar{y}_{t+1} - \mathbf{1}\bar{y}_t\|,
\end{aligned}
$$

where the second inequality is because it holds that $\left\|\mathbf{x} - \frac{1}{m}\mathbf{1}\mathbf{1}^\top \mathbf{x}\right\| \leq \|\mathbf{x}\|$ for any $\mathbf{x} \in \mathbb{R}^{m \times d}$ from [14].

Then we only need to consider the term $\|\mathbf{1}\bar{y}_{t+1} - \mathbf{1}\bar{y}_t\|$. Using the iteration of average variables illustrated in Eqn. (18), we have

$$
\begin{aligned}
& \|\mathbf{1}\bar{y}_{t+1} - \mathbf{1}\bar{y}_t\| \\
= & \left\| \frac{2}{1+\alpha}\mathbf{1}\bar{x}_{t+1} - \frac{1-\alpha}{1+\alpha}\mathbf{1}\bar{x}_t - \mathbf{1}\bar{y}_t \right\| \\
= & \left\| \frac{2}{1+\alpha}\cdot\frac{1}{m}\mathbf{1}\mathbf{1}^\top \left(\mathbf{prox}_{\eta m, R}(\mathbf{y}_t - \eta \mathbf{s}_t)\right) - \frac{1-\alpha}{1+\alpha}\cdot\frac{1}{m}\mathbf{1}\mathbf{1}^\top \mathbf{x}_t - \frac{1}{m}\mathbf{1}\mathbf{1}^\top \mathbf{y}_t \right\| \\
= & \left\| \frac{2}{1+\alpha}\cdot\frac{1}{m}\mathbf{1}\mathbf{1}^\top \left(\mathbf{prox}_{\eta m, R}(\mathbf{y}_t - \eta \mathbf{s}_t) - \mathbf{y}_t\right) - \frac{1-\alpha}{1+\alpha}\mathbf{1}\left(\bar{x}_t - \bar{y}_t\right) \right\| \\
\leq & \frac{2}{1+\alpha}\cdot \left\| \frac{1}{m}\mathbf{1}\mathbf{1}^\top \left(\mathbf{y}_t - \mathbf{prox}_{\eta m, R}(\mathbf{y}_t - \eta \mathbf{s}_t)\right) - \eta\mathbf{1}\cdot\tilde{\nabla}h(\bar{y}_t) \right\| \\
& + \frac{2}{1+\alpha}\left\| \eta\mathbf{1}\cdot\tilde{\nabla}h(\bar{y}_t) \right\| + \frac{1-\alpha}{1+\alpha}\left(\|\mathbf{1}\bar{x}_t - \mathbf{1}x^*\| + \|\mathbf{1}\bar{y}_t - \mathbf{1}x^*\|\right).
\end{aligned}
$$

Furthermore, by Lemma 12, we can obtain that

$$
\frac{2}{1+\alpha}\cdot \left\| \frac{1}{m}\mathbf{1}\mathbf{1}^\top \left(\mathbf{y}_t - \mathbf{prox}_{\eta m, R}(\mathbf{y}_t - \eta \mathbf{s}_t)\right) - \eta\mathbf{1}\cdot\tilde{\nabla}h(\bar{y}_t) \right\| \leq (8+4M\eta) \|\mathbf{y}_t - \mathbf{1}\bar{y}_t\| + 4\eta \|\mathbf{s}_t - \mathbf{1}\bar{s}_t\|.
$$

Furthermore, by Lemma 8 and the fact that $\tilde{\nabla}h(x^*) = 0$, we can obtain

$$\frac{2}{1+\alpha}\left\|\eta\mathbf{1}\cdot\tilde{\nabla}h(\bar{y}_t)\right\| + \frac{1-\alpha}{1+\alpha}\left(\|\mathbf{1}\bar{x}_t - \mathbf{1}x^*\| + \|\mathbf{1}\bar{y}_t - \mathbf{1}x^*\|\right)$$

$$\leq\frac{2\eta\sqrt{m}}{1+\alpha}\left\|\tilde{\nabla}h(\bar{y}_t) - \tilde{\nabla}h(x^*)\right\| + \frac{1-\alpha}{1+\alpha}\cdot 2\cdot\sqrt{\frac{2}{\mu}V_t}$$

$$\leq\left(\frac{6\eta\sqrt{m}L}{1+\alpha} + \frac{1-\alpha}{1+\alpha}\cdot 2\right)\sqrt{\frac{2}{\mu}V_t}$$

$$\leq\left(6\sqrt{m} + 2\right)\sqrt{\frac{2}{\mu}V_t},$$

where the second inequality is because of the Eqn.(15). Thus, we can obtain that

$$\|\mathbf{1}\bar{y}_{t+1} - \mathbf{1}\bar{y}_t\| \leq (8 + 4M\eta)\|\mathbf{y}_t - \mathbf{1}\bar{y}_t\| + 4\eta\|\mathbf{s}_t - \mathbf{1}\bar{s}_t\| + \left(6\sqrt{m} + 2\right)\sqrt{\frac{2}{\mu}V_t}.$$

Combining above results, we can bound the value of $\|\mathbf{s}_{t+1} - \mathbf{1}\bar{s}_{t+1}\|$ as follows

$$\|\mathbf{s}_{t+1} - \mathbf{1}\bar{s}_{t+1}\| \leq \left(\rho + 4\rho^3 M\eta\right)\|\mathbf{s}_t - \mathbf{1}\bar{s}_t\| + 2\rho^2 M\|\mathbf{x}_t - \mathbf{1}\bar{x}_t\| + \left(\rho M + 4\rho^3 M\right)\|\mathbf{1}\bar{y}_t - \mathbf{y}_t\|$$

$$+ \rho M\|\mathbf{1}\bar{y}_{t+1} - \mathbf{1}\bar{y}_t\|$$

$$\leq \left(\rho + 4\rho^3 M\eta\right)\|\mathbf{s}_t - \mathbf{1}\bar{s}_t\| + 2\rho^2 M\|\mathbf{x}_t - \mathbf{1}\bar{x}_t\| + \left(\rho M + 4\rho^3 M\right)\|\mathbf{1}\bar{y}_t - \mathbf{y}_t\|$$

$$+ \rho M\left((8 + 4M\eta)\|\mathbf{y}_t - \mathbf{1}\bar{y}_t\| + 4\eta\|\mathbf{s}_t - \mathbf{1}\bar{s}_t\| + \left(6\sqrt{m} + 2\right)\sqrt{\frac{2}{\mu}V_t}\right)$$

$$\leq \left(\rho + 4\rho^3 M\eta + 4\rho M\eta\right)\|\mathbf{s}_t - \mathbf{1}\bar{s}_t\| + 2\rho^2 M\|\mathbf{x}_t - \mathbf{1}\bar{x}_t\|$$

$$+ \left(9\rho M + 4\rho^3 M + 4\rho M^2\eta\right)\|\mathbf{1}\bar{y}_t - \mathbf{y}_t\| + 8\rho M\sqrt{m}\sqrt{\frac{2}{\mu}V_t}.$$

If we denote $\mathbf{z}_t = [\|\mathbf{x}_t - \mathbf{1}\bar{x}_t\|, \|\mathbf{y}_t - \mathbf{1}\bar{y}_t\|, \|\mathbf{s}_t - \mathbf{1}\bar{s}_t\|]^\top$, then due to $\rho < 1$, $L \leq M$, and $1 \leq m$, we can have

$$\mathbf{z}_{t+1} \leq \mathbf{A}'\mathbf{z}_t + 8\rho M\sqrt{m}[0, 0, \sqrt{\frac{2}{\mu}V_t}]^\top$$

where

$$\mathbf{A}' = \begin{pmatrix} 0 & 2\rho & 2\rho\eta \\ 2\rho & 4\rho^2 & 4\rho^2\eta \\ 2\rho^2 M & 9\rho M + 4\rho^3 M + 4\rho M^2\eta & \rho + 4\rho^3 M\eta + 4\rho M\eta \end{pmatrix}$$

$$\leq \begin{pmatrix} 0 & 2\rho & 2\rho\eta \\ 2\rho & 4\rho & 4\rho\eta \\ 2\rho M & \rho M(13 + 4M\eta) & \rho(1 + 6M\eta) \end{pmatrix}$$

$$= \rho\begin{pmatrix} 0 & 2 & 2\eta \\ 2 & 4 & 4\eta \\ 2M & M(13 + 4M\eta) & 1 + 6M\eta \end{pmatrix} = \mathbf{A}$$

Finally, we simplify the result and obtain that

$$\mathbf{z}_{t+1} \leq \mathbf{A}\mathbf{z}_t + 8\rho M\sqrt{m}[0, 0, \sqrt{\frac{2}{\mu}V_t}]^\top.$$

$\square$

## C.2 Proof of Lemma 2

*Proof.* By the $\mu$-strongly convexity , $L$-smoothness of $f(x)$ and the property of proximal operator showed in Lemma 9, we can have for any given $z \in \mathbb{R}^d$

$$
\begin{aligned}
&h(\mathbf{prox}_{\eta,r}(\mathbf{y}_t^{(i)} - \eta\mathbf{s}_t^{(i)})) = f(\mathbf{prox}_{\eta,r}(\mathbf{y}_t^{(i)} - \eta\mathbf{s}_t^{(i)})) + r(\mathbf{prox}_{\eta,r}(\mathbf{y}_t^{(i)} - \eta\mathbf{s}_t^{(i)})) \\
&= f(\mathbf{y}_t^{(i)} + \mathbf{prox}_{\eta,r}(\mathbf{y}_t^{(i)} - \eta\mathbf{s}_t^{(i)}) - \mathbf{y}_t^{(i)}) + r(\mathbf{prox}_{\eta,r}(\mathbf{y}_t^{(i)} - \eta\mathbf{s}_t^{(i)})) \\
&\leq f(\mathbf{y}_t^{(i)}) + \nabla f(\mathbf{y}_t^{(i)})^\top \left( \mathbf{prox}_{\eta,r}(\mathbf{y}_t^{(i)} - \eta\mathbf{s}_t^{(i)}) - \mathbf{y}_t^{(i)} \right) + \frac{L}{2} \left\| \mathbf{prox}_{\eta,r}(\mathbf{y}_t^{(i)} - \eta\mathbf{s}_t^{(i)}) - \mathbf{y}_t^{(i)} \right\|^2 \\
&\quad + r(z) + \frac{1}{\eta}(\mathbf{prox}_{\eta,r}(\mathbf{y}_t^{(i)} - \eta\mathbf{s}_t^{(i)}) - \mathbf{y}_t^{(i)} + \eta\mathbf{s}_t^{(i)})^\top (z - \mathbf{prox}_{\eta,r}(\mathbf{y}_t^{(i)} - \eta\mathbf{s}_t^{(i)})) \\
&\leq h(z) - \nabla f(\mathbf{y}_t^{(i)})^\top (z - \mathbf{y}_t^{(i)}) - \frac{\mu}{2} \left\| z - \mathbf{y}_t^{(i)} \right\|^2 + \nabla f(\mathbf{y}_t^{(i)})^\top \left( \mathbf{prox}_{\eta,r}(\mathbf{y}_t^{(i)} - \eta\mathbf{s}_t^{(i)}) - \mathbf{y}_t^{(i)} \right) \\
&\quad + \frac{L}{2} \left\| \mathbf{prox}_{\eta,r}(\mathbf{y}_t^{(i)} - \eta\mathbf{s}_t^{(i)}) - \mathbf{y}_t^{(i)} \right\|^2 + \frac{1}{\eta}(\mathbf{prox}_{\eta,r}(\mathbf{y}_t^{(i)} - \eta\mathbf{s}_t^{(i)}) - \mathbf{y}_t^{(i)} + \eta\mathbf{s}_t^{(i)})^\top (z - \mathbf{prox}_{\eta,r}(\mathbf{y}_t^{(i)} - \eta\mathbf{s}_t^{(i)})).
\end{aligned}
\tag{21}
$$

Multiplying $1 - \alpha$ on both sides of Eqn. (21) and setting $z = \bar{x}_t$, we get

$$
\begin{aligned}
&(1 - \alpha)h(\mathbf{prox}_{\eta,r}(\mathbf{y}_t^{(i)} - \eta\mathbf{s}_t^{(i)})) \\
&\leq (1 - \alpha)h(\bar{x}_t) - (1 - \alpha)\nabla f(\mathbf{y}_t^{(i)})^\top (\bar{x}_t - \mathbf{y}_t^{(i)}) - \frac{\mu(1 - \alpha)}{2} \left\| \bar{x}_t - \mathbf{y}_t^{(i)} \right\|^2 \\
&\quad + (1 - \alpha)\nabla f(\mathbf{y}_t^{(i)})^\top \left( \mathbf{prox}_{\eta,r}(\mathbf{y}_t^{(i)} - \eta\mathbf{s}_t^{(i)}) - \mathbf{y}_t^{(i)} \right) + \frac{L(1 - \alpha)}{2} \left\| \mathbf{prox}_{\eta,r}(\mathbf{y}_t^{(i)} - \eta\mathbf{s}_t^{(i)}) - \mathbf{y}_t^{(i)} \right\|^2 \\
&\quad + \frac{(1 - \alpha)}{\eta}(\mathbf{prox}_{\eta,r}(\mathbf{y}_t^{(i)} - \eta\mathbf{s}_t^{(i)}) - \mathbf{y}_t^{(i)} + \eta\mathbf{s}_t^{(i)})^\top (\bar{x}_t - \mathbf{prox}_{\eta,r}(\mathbf{y}_t^{(i)} - \eta\mathbf{s}_t^{(i)})).
\end{aligned}
\tag{22}
$$

Similarly multiplying $\alpha$ on both sides of Eqn. (21) and setting $z = x^*$, we obtain that

$$
\begin{aligned}
&\alpha(\mathbf{prox}_{\eta,r}(\mathbf{y}_t^{(i)} - \eta\mathbf{s}_t^{(i)})) \\
&\leq \alpha h(x^*) - \alpha\nabla f(\mathbf{y}_t^{(i)})^\top (x^* - \mathbf{y}_t^{(i)}) - \frac{\mu\alpha}{2} \left\| x^* - \mathbf{y}_t^{(i)} \right\|^2 \\
&\quad + \alpha\nabla f(\mathbf{y}_t^{(i)})^\top \left( \mathbf{prox}_{\eta,r}(\mathbf{y}_t^{(i)} - \eta\mathbf{s}_t^{(i)}) - \mathbf{y}_t^{(i)} \right) + \frac{\alpha L}{2} \left\| \mathbf{prox}_{\eta,r}(\mathbf{y}_t^{(i)} - \eta\mathbf{s}_t^{(i)}) - \mathbf{y}_t^{(i)} \right\|^2 \\
&\quad + \frac{\alpha}{\eta}(\mathbf{prox}_{\eta,r}(\mathbf{y}_t^{(i)} - \eta\mathbf{s}_t^{(i)}) - \mathbf{y}_t^{(i)} + \eta\mathbf{s}_t^{(i)})^\top (x^* - \mathbf{prox}_{\eta,r}(\mathbf{y}_t^{(i)} - \eta\mathbf{s}_t^{(i)})).
\end{aligned}
\tag{23}
$$

Adding above two inequalities and by the definition of $G_t$ and $G_t^{(i)}$ in Lemma 12, we have

$$
\begin{aligned}
&h(\mathbf{prox}_{\eta,r}(\mathbf{y}_t^{(i)} - \eta\mathbf{s}_t^{(i)})) - h(x^*) \\
&\leq (1 - \alpha) \left( h(\bar{x}_t) - h(x^*) \right) - \alpha \cdot \frac{\mu}{2} \left\| x^* - \mathbf{y}_t^{(i)} \right\|^2 + \frac{1}{2L} \left\| G_t^{(i)} \right\|^2 \\
&\quad + \left( \mathbf{s}_t^{(i)} - G_t^{(i)} - \nabla f(\mathbf{y}_t^{(i)}) \right)^\top \left( (1 - \alpha)\bar{x}_t + \alpha x^* + \eta G_t^{(i)} - \mathbf{y}_t^{(i)} \right).
\end{aligned}
\tag{24}
$$

Note that by Jensen's inequality, we can get that

$$
\| x^* - \bar{y}_t \| = \left\| x^* - \frac{1}{m}\sum_{i=1}^m \mathbf{y}_t^{(i)} \right\| \leq \sqrt{\frac{1}{m}\sum_{i=1}^m \left\| x^* - \mathbf{y}_t^{(i)} \right\|^2}.
$$

Then averaging Eqn. (24) from $i = 1$ to $i = m$, we have

$$
\begin{aligned}
& h(\bar{x}_{t+1}) - h(x^*) \\
\leq & \frac{1}{m} \sum_{i=1}^{m} h(\mathbf{prox}_{\eta,r}(\mathbf{y}_t^{(i)} - \eta \mathbf{s}_t^{(i)})) - h(x^*) \\
\leq & (1-\alpha)\left(h(\bar{x}_t) - h(x^*)\right) + \frac{1}{m} \sum_{i=1}^{m} \frac{1}{2L} \left\| G_t^{(i)} \right\|^2 \\
& + \frac{1}{m} \sum_{i=1}^{m} \left(\mathbf{s}_t^{(i)} - G_t^{(i)} - \nabla f(\mathbf{y}_t^{(i)})\right)^\top \left((1-\alpha)\bar{x}_t + \alpha x^* + \eta G_t^{(i)} - \mathbf{y}_t^{(i)}\right) - \alpha \cdot \frac{\mu}{2} \|x^* - \bar{y}_t\|^2 \\
= & (1-\alpha)\left(h(\bar{x}_t) - h(x^*)\right) - \frac{1}{m} \sum_{i=1}^{m} \frac{1}{2L} \left\| G_t^{(i)} \right\|^2 + \frac{1}{m} \sum_{i=1}^{m} \left(\mathbf{s}_t^{(i)} - \nabla f(\mathbf{y}_t^{(i)})\right)^\top \left((1-\alpha)\bar{x}_t + \alpha x^* - \bar{y}_t\right) \\
& - \frac{1}{m} \sum_{i=1}^{m} (G_t^{(i)} - \tilde{\nabla} h(\bar{y}_t))^\top \left((1-\alpha)\bar{x}_t + \alpha x^* - \bar{y}_t\right) + \frac{\eta}{m} \sum_{i=1}^{m} \left(\mathbf{s}_t^{(i)} - \nabla f(\mathbf{y}_t^{(i)})\right)^\top \left(G_t^{(i)} - \tilde{\nabla} h(\bar{y}_t)\right) \\
& + \frac{1}{m} \sum_{i=1}^{m} \left(\mathbf{s}_t^{(i)} - \nabla f(\mathbf{y}_t^{(i)})\right)^\top \left(\bar{y}_t - \mathbf{y}_t^{(i)}\right) - \frac{1}{m} \sum_{i=1}^{m} (G_t^{(i)} - \tilde{\nabla} h(\bar{y}_t))^\top \left(\bar{y}_t - \mathbf{y}_t^{(i)}\right) \\
& - \frac{1}{m} \sum_{i=1}^{m} \tilde{\nabla} h(\bar{y}_t)^\top \left(\left((1-\alpha)\bar{x}_t + \alpha x^* - \bar{y}_t\right) - \eta \left(\mathbf{s}_t^{(i)} - \nabla f(\mathbf{y}_t^{(i)})\right) + \left(\bar{y}_t - \mathbf{y}_t^{(i)}\right)\right) - \alpha \cdot \frac{\mu}{2} \|x^* - \bar{y}_t\|^2,
\end{aligned}
$$

(25)

where the first inequality is because of the convexity of $h(x)$, the second inequality is from Eqn. (24), and the last equality is a reorganization of above expression.

For $\left\| G_t^{(i)} \right\|^2$, we have following equality

$$
\begin{aligned}
\left\| G_t^{(i)} \right\|^2 &= \left\| G_t^{(i)} - \tilde{\nabla} h(\bar{y}_t) + \tilde{\nabla} h(\bar{y}_t) \right\|^2 \\
&= \left\| G_t^{(i)} - \tilde{\nabla} h(\bar{y}_t) \right\|^2 + 2 \left\langle G_t^{(i)} - \tilde{\nabla} h(\bar{y}_t), \tilde{\nabla} h(\bar{y}_t) \right\rangle + \left\| \tilde{\nabla} h(\bar{y}_t) \right\|^2 .
\end{aligned}
$$

(26)

Now we bound another term of $V_{t+1}$ as follows

$$
\begin{aligned}
& \frac{\mu}{2} \|\bar{v}_{t+1} - x^*\|^2 \\
= & \frac{\mu}{2} \left\| \bar{x}_t + \frac{1}{\alpha}(\bar{x}_{t+1} - \bar{x}_t) - x^* \right\|^2 \\
= & \frac{\mu}{2} \left\| \bar{x}_t + \frac{1}{\alpha}(\frac{1}{m} \mathbf{1}^\top \mathbf{prox}_{\eta m, R}(\mathbf{y}_t - \eta \mathbf{s}_t) - \bar{x}_t) - x^* \right\|^2 \\
\leq & \frac{\mu}{2} \left\| \bar{x}_t + \frac{1}{\alpha}(\bar{y}_t - \eta \tilde{\nabla} h(\bar{y}_t) - \bar{x}_t) - x^* \right\|^2 + \frac{\mu}{2} \left\| \bar{y}_t - \eta \tilde{\nabla} h(\bar{y}_t) - \frac{1}{m} \mathbf{1}^\top \mathbf{prox}_{\eta m, R}(\mathbf{y}_t - \eta \mathbf{s}_t) \right\|^2 \\
& + \mu \left\| \bar{x}_t + \frac{1}{\alpha}(\bar{y}_t - \eta \tilde{\nabla} h(\bar{y}_t) - \bar{x}_t) - x^* \right\| \cdot \left\| \bar{y}_t - \eta \tilde{\nabla} h(\bar{y}_t) - \frac{1}{m} \mathbf{1}^\top \mathbf{prox}_{\eta m, R}(\mathbf{y}_t - \eta \mathbf{s}_t) \right\|.
\end{aligned}
$$

Because of $\bar{y}_t - \bar{x}_t = \alpha(\bar{y}_t - \bar{v}_t)$, we have

$$
\begin{aligned}
\bar{x}_t + \frac{1}{\alpha}(\bar{y}_t - \eta \tilde{\nabla} h(\bar{y}_t) - \bar{x}_t) - x^* &= \bar{v}_t - \alpha(\bar{v}_t - \bar{y}_t) - \frac{\eta}{\alpha} \tilde{\nabla} h(\bar{y}_t) - x^* \\
&= (1-\alpha)(\bar{v}_t - x^*) + \alpha(\bar{y}_t - x^*) - \frac{\eta}{\alpha} \tilde{\nabla} h(\bar{y}_t).
\end{aligned}
$$

Thus, we can obtain

$$\frac{\mu}{2}\left\|\bar{x}_t + \frac{1}{\alpha}(\bar{y}_t - \eta\tilde{\nabla}h(\bar{y}_t) - \bar{x}_t) - x^*\right\|^2$$

$$=\frac{\mu - \mu\alpha}{2}\|\bar{v}_t - x^*\|^2 + \frac{\mu\alpha}{2}\|\bar{y}_t - x^*\|^2 - \alpha\left\langle\tilde{\nabla}h(\bar{y}_t), (1-\alpha)\bar{v}_t + \alpha\bar{y}_t - x^*\right\rangle$$

$$+ \frac{1}{2L}\left\|\tilde{\nabla}h(\bar{y}_t)\right\|^2 - \frac{\alpha(\mu - \mu\alpha)}{2}\|\bar{y}_t - \bar{v}_t\|^2$$

$$\leq\frac{\mu - \mu\alpha}{2}\|\bar{v}_t - x^*\|^2 + \frac{\mu\alpha}{2}\|\bar{y}_t - x^*\|^2 - \alpha\left\langle\tilde{\nabla}h(\bar{y}_t), (1-\alpha)\bar{v}_t + \alpha\bar{y}_t - x^*\right\rangle + \frac{1}{2L}\left\|\tilde{\nabla}h(\bar{y}_t)\right\|^2$$

$$=\frac{\mu - \mu\alpha}{2}\|\bar{v}_t - x^*\|^2 + \frac{\mu\alpha}{2}\|\bar{y}_t - x^*\|^2 + \left\langle\tilde{\nabla}h(\bar{y}_t), \alpha x^* + (1-\alpha)\bar{x}_t - \bar{y}_t\right\rangle + \frac{1}{2L}\left\|\tilde{\nabla}h(\bar{y}_t)\right\|^2,$$

where the last equality comes from Lemma 7. We can further obtain that

$$\frac{\mu}{2}\|\bar{v}_{t+1} - x^*\|^2$$

$$=\frac{\mu}{2}\left\|\bar{x}_t + \frac{1}{\alpha}(\bar{y}_t - \eta\tilde{\nabla}h(\bar{y}_t) - \bar{x}_t) - x^*\right\|^2 + \frac{\mu}{2}\left\|\bar{y}_t - \eta\tilde{\nabla}h(\bar{y}_t) - \frac{1}{m}\mathbf{1}^\top\mathbf{prox}_{\eta m, R}(\mathbf{y}_t - \eta\mathbf{s}_t)\right\|^2$$

$$+ \frac{\mu}{2}\left\langle\bar{x}_t + \frac{1}{\alpha}(\bar{y}_t - \eta\tilde{\nabla}h(\bar{y}_t) - \bar{x}_t) - x^*, \bar{y}_t - \eta\tilde{\nabla}h(\bar{y}_t) - \frac{1}{m}\mathbf{1}^\top\mathbf{prox}_{\eta m, R}(\mathbf{y}_t - \eta\mathbf{s}_t)\right\rangle$$

$$\leq\frac{\mu - \mu\alpha}{2}\|\bar{v}_t - x^*\|^2 + \frac{\mu\alpha}{2}\|\bar{y}_t - x^*\|^2 + \left\langle\tilde{\nabla}h(\bar{y}_t), \alpha x^* + (1-\alpha)\bar{x}_t - \bar{y}_t\right\rangle + \frac{1}{2L}\left\|\tilde{\nabla}h(\bar{y}_t)\right\|^2$$

$$+ \frac{\mu}{2}\left\|\frac{\eta}{m}\mathbf{1}^\top G_t - \eta\tilde{\nabla}h(\bar{y}_t)\right\|^2 + \frac{\mu}{2}\left\|\bar{x}_t + \frac{1}{\alpha}(\bar{y}_t - \eta\tilde{\nabla}h(\bar{y}_t) - \bar{x}_t) - x^*\right\| \cdot \left\|\frac{\eta}{m}\mathbf{1}^\top G_t - \eta\tilde{\nabla}h(\bar{y}_t)\right\|.$$

$$(27)$$

Recalling for any $a > 0, b > 0, \{a_i > 0\}$, it holds that $ab \leq \frac{a^2 + b^2}{2}$, $\frac{1}{m}\sum_{i=1}^m a_i \leq \sqrt{\frac{1}{m}\sum_{i=1}^m a_i^2}$, and $(a+b)^2 \leq 2a^2 + 2b^2$. Combining with Cauchy's inequality, Eqn. (25), Eqn. (26) and Eqn. (27), we can obtain that

$$V_{t+1}$$

$$=h(\bar{x}_t) - h(x^*) + \frac{\mu}{2}\|\bar{x}_t - x^*\|^2$$

$$\leq(1-\alpha)V_t + \left\|\frac{\eta}{m}\mathbf{1}^\top G_t - \eta\tilde{\nabla}h(\bar{y}_t)\right\| \cdot \left\|\tilde{\nabla}h(\bar{y}_t)\right\|$$

$$+ \sqrt{\frac{1}{m}\sum_{i=1}^m\left\|\mathbf{s}_t^{(i)} - \nabla f(\mathbf{y}_t^{(i)})\right\|^2} \cdot \|(1-\alpha)\bar{x}_t + \alpha x^* - \bar{y}_t\|$$

$$+ \left\|\frac{1}{m}\mathbf{1}^\top G_t - \tilde{\nabla}h(\bar{y}_t)\right\| \cdot \|(1-\alpha)\bar{x}_t + \alpha x^* - \bar{y}_t\| + \frac{2\eta}{m}\sum_{i=1}^m\left\|\mathbf{s}_t^{(i)} - \nabla f(\mathbf{y}_t^{(i)})\right\|^2$$

$$+ \frac{2}{m}\sum_{i=1}^m\left\|\mathbf{s}_t^{(i)} - \nabla f(\mathbf{y}_t^{(i)})\right\|^2 + \frac{4}{m}\|\mathbf{1}\bar{y}_t - \mathbf{y}_t\|^2 + \frac{2}{m}\left\|G_t - \mathbf{1}\tilde{\nabla}h(\bar{y}_t)\right\|^2 + \frac{2\eta}{m}\left\|G_t - \mathbf{1}\tilde{\nabla}h(\bar{y}_t)\right\|^2$$

$$+ \left\|\tilde{\nabla}h(\bar{y}_t)\right\|\left(\eta\sqrt{\frac{1}{m}\sum_{i=1}^m\left\|\mathbf{s}_t^{(i)} - \nabla f(\mathbf{y}_t^{(i)})\right\|^2}\right) + \frac{\mu}{2}\left\|\frac{\eta}{m}\mathbf{1}^\top G_t - \eta\tilde{\nabla}h(\bar{y}_t)\right\|^2$$

$$+ \frac{\mu}{2}\left((1-\alpha)\|\bar{v}_t - x^*\| + \alpha\|\bar{y}_t - x^*\| + \frac{\eta}{\alpha}\left\|\tilde{\nabla}h(\bar{y}_t)\right\|\right) \cdot \left\|\frac{\eta}{m}\mathbf{1}^\top G_t - \eta\tilde{\nabla}h(\bar{y}_t)\right\|.$$

$$(28)$$

By the lemma 8, 10, 11, 12, and 13, we can collect the following inequalities

$$\left\|G_t - \mathbf{1}\tilde{\nabla}h(\bar{y}_t)\right\|^2 \leq 2\left(2M+7L\right)^2 \|\mathbf{y}_t - \mathbf{1}\bar{y}_t\|^2 + 32\eta^2 \|\mathbf{s}_t - \mathbf{1}\bar{s}_t\|^2$$

$$\left\|\frac{\eta}{m}\mathbf{1}^\top G_t - \eta\tilde{\nabla}h(\bar{y}_t)\right\| \leq \frac{4+2M\eta}{\sqrt{m}}\|\mathbf{y}_t - \mathbf{1}\bar{y}_t\| + \frac{2\eta}{\sqrt{m}}\|\mathbf{s}_t - \mathbf{1}\bar{s}_t\|$$

$$\sum_{i=1}^m \left\|\mathbf{s}_t^{(i)} - \nabla f(\mathbf{y}_t^{(i)})\right\|^2 \leq 2\|\mathbf{s}_t - \mathbf{1}\bar{s}_t\|^2 + 8M^2\|\mathbf{1}\bar{x}_t - \mathbf{x}_t\|^2$$

$$\left\|\tilde{\nabla}h(\bar{y}_t)\right\| \leq 3L\sqrt{\frac{2}{\mu}V_t}$$

$$\|(1-\alpha)\bar{x}_t + \alpha x^* - \bar{y}_t\| \leq \sqrt{\frac{2}{\mu}V_t}.$$

Replacing above inequalities to Eqn. (28), we can obtain

$$V_{t+1}$$
$$\leq (1-\alpha)V_t + \left(\frac{4+2M\eta}{\sqrt{m}}\|\mathbf{y}_t - \mathbf{1}\bar{y}_t\| + \frac{2\eta}{\sqrt{m}}\|\mathbf{s}_t - \mathbf{1}\bar{s}_t\|\right)\cdot\left(3L\sqrt{\frac{2}{\mu}V_t}\right)$$
$$+ \frac{1}{\sqrt{m}}\cdot\sqrt{2\|\mathbf{s}_t - \mathbf{1}\bar{s}_t\|^2 + 8M^2\|\mathbf{1}\bar{x}_t - \mathbf{x}_t\|^2}\cdot\sqrt{\frac{2}{\mu}V_t}$$
$$+ L\left(\frac{4+2M\eta}{\sqrt{m}}\|\mathbf{y}_t - \mathbf{1}\bar{y}_t\| + \frac{2\eta}{\sqrt{m}}\|\mathbf{s}_t - \mathbf{1}\bar{s}_t\|\right)\cdot\sqrt{\frac{2}{\mu}V_t} + \frac{2(1+\eta)}{m}\left(2\|\mathbf{s}_t - \mathbf{1}\bar{s}_t\|^2 + 8M^2\|\mathbf{1}\bar{x}_t - \mathbf{x}_t\|^2\right)$$
$$+ \frac{4}{m}\|\mathbf{1}\bar{y}_t - \mathbf{y}_t\|^2 + \frac{2(1+\eta)}{m}\left(2\left(2M+7L\right)^2\|\mathbf{y}_t - \mathbf{1}\bar{y}_t\|^2 + 32\eta^2\|\mathbf{s}_t - \mathbf{1}\bar{s}_t\|^2\right)$$
$$+ 3\sqrt{\frac{2}{\mu}V_t}\cdot\sqrt{\frac{1}{m}\left(2\|\mathbf{s}_t - \mathbf{1}\bar{s}_t\|^2 + 8M^2\|\mathbf{1}\bar{x}_t - \mathbf{x}_t\|^2\right)} + \frac{\mu}{2}\left(\frac{4+2M\eta}{\sqrt{m}}\|\mathbf{y}_t - \mathbf{1}\bar{y}_t\| + \frac{2\eta}{\sqrt{m}}\|\mathbf{s}_t - \mathbf{1}\bar{s}_t\|\right)^2$$
$$+ \frac{\mu}{2}\left(\sqrt{\frac{2}{\mu}V_t} + \frac{\eta}{\alpha}\cdot 3L\sqrt{\frac{2}{\mu}V_t}\right)\cdot\left(\frac{4+2M\eta}{\sqrt{m}}\|\mathbf{y}_t - \mathbf{1}\bar{y}_t\| + \frac{2\eta}{\sqrt{m}}\|\mathbf{s}_t - \mathbf{1}\bar{s}_t\|\right).$$
$$\leq (1-\alpha)V_t + \left(\frac{4(4+2M\eta)}{\sqrt{m}\eta} + \frac{\mu}{2}(\frac{3}{\alpha}+1)\cdot\frac{4+2M\eta}{\sqrt{m}}\right)\sqrt{\frac{2}{\mu}V_t}\cdot\|\mathbf{y}_t - \mathbf{1}\bar{y}_t\|$$
$$+ \frac{8\sqrt{2}L}{\sqrt{m}}\cdot\sqrt{\frac{2}{\mu}V_t}\cdot\|\mathbf{x}_t - \mathbf{1}\bar{x}_t\| + \left(\frac{8}{\sqrt{m}} + \frac{4\sqrt{2}}{\sqrt{m}} + \frac{\mu}{2}(\frac{3}{\alpha}+1)\cdot\frac{2\eta}{\sqrt{m}}\right)\sqrt{\frac{2}{\mu}V_t}\cdot\|\mathbf{s}_t - \mathbf{1}\bar{s}_t\|$$
$$+ \frac{(2\mu+68+68\eta)\,\eta^2}{m}\|\mathbf{s}_t - \mathbf{1}\bar{s}_t\|^2 + \frac{16\,(\eta+1)\,L^2}{m}\|\mathbf{x}_t - \mathbf{1}\bar{x}_t\|^2$$
$$+ \left(\frac{4}{m} + \frac{\mu(4+2M\eta)^2}{m} + \frac{4(1+\eta)(2M+7L)^2}{m}\right)\|\mathbf{y}_t - \mathbf{1}\bar{y}_t\|^2$$
$$\leq (1-\alpha)V_t + \frac{4}{\sqrt{m}}\left(21L + 2\sqrt{L\mu}(2+2M\eta) + 3\mu(2+2M\eta) + 8 + \frac{\mu}{L} + 3\sqrt{\frac{\mu}{L}}\right)\cdot\|\mathbf{z}_t\|\sqrt{\frac{2}{\mu}V_t}$$
$$+ \frac{8}{m}(1+\mu)\cdot\left(9\cdot(12+8L+4M)^2 + 16L(L+1) + \frac{133+79L}{L}\right)\|\mathbf{z}_t\|^2,$$

where the last inequality is because of Cauchy's inequality and $L \leq M$.  $\square$

### C.3  Proof of Lemma 3

*Proof.* It is easy to check that spectral norm of a matrix with all the indices positive will be less that the sum of all its indices, then we have $\|\mathbf{A}\|_2 \leq \|\mathbf{A}\|_F \leq \rho D_3$, where $\|\mathbf{A}\|_F$ is the Frobenius norm of $\mathbf{A}$ and
$$D_3 = \sum_{i,j}\mathbf{A}_{i,j}/\rho = (9 + 6\eta + 15M + 4M^2\eta + 6M\eta).$$

Therefore, if $\rho < \frac{1}{2D_3}$, we will have $\|\mathbf{A}\|_2 < \frac{1}{2}$. $\qquad\square$

## C.4 Proof of Lemma 4

Our proof is based on the fact that error term $\|\mathbf{z}_t\|$ in Eqn. (8) can be controlled by $\rho$. In total, what we will do in this lemma are all about bounding the term $D_1\sqrt{V_t}\cdot\|\mathbf{z}_t\|$ and $D_2\|\mathbf{z}_t\|^2$ to the level of $\frac{\alpha}{2}V_t$ in an induction way.

*Proof.* We will prove our result by induction. First, when $t = 0$, by Lemma 2 and the definition of $C$, we have

$$
\begin{aligned}
V_1 &\leq (1-\alpha)V_0 + D_1\|\mathbf{z}_0\|\sqrt{V_0} + D_2\|\mathbf{z}_0\|^2 \\
&\leq (1-\alpha)V_0 + \frac{1}{2}\alpha V_0 + \frac{D_1^2}{2\alpha}\cdot\|\mathbf{z}_0\|^2 + D_2\|\mathbf{z}_0\|^2 \\
&\leq \left(1-\frac{\alpha}{2}\right)\left(V_0 + 2\left(\frac{D_1^2}{2\alpha}+D_2\right)\|\mathbf{z}_0\|^2\right) \\
&= \left(1-\frac{\alpha}{2}\right)\left(V_0 + C\|\mathbf{z}_0\|^2\right).
\end{aligned}
$$

Therefore, we have that Eqn. (9) holds for $t = 1$. Next, we assume that for $i = 1,\ldots,t$, it holds that

$$
V_t \leq \left(1-\frac{\alpha}{2}\right)^t\left(V_0 + C\|\mathbf{z}_0\|^2\right).
$$

By Lemma 3, we known that the spectral norm of $\mathbf{A}$ is upper bounded by $\frac{1}{2}$ and $\rho D_3$. Then we can obtain that

$$
\begin{aligned}
\|\mathbf{z}_t\| &\leq 8M\rho\sqrt{\frac{2m}{\mu}}\sum_{j=0}^{t-1}2^{-(t-1-j)}\sqrt{V_j} + 2^{-(t-1)}\cdot\rho D_3\|\mathbf{z}_0\| \\
&\leq 8M\rho\sqrt{\frac{2m}{\mu}}\sum_{j=0}^{t-1}2^{-(t-1-j)}\left(\sqrt{1-\frac{\alpha}{2}}\right)^j\left(\sqrt{V_0}+\sqrt{C}\|\mathbf{z}_0\|\right) + 2^{-t}\cdot(2\rho D_3\|\mathbf{z}_0\|) \\
&= 8M\rho\sqrt{\frac{2m}{\mu}}\frac{2\left(\sqrt{1-\frac{\alpha}{2}}\right)^t - 2^{-(t-1)}}{2\sqrt{1-\frac{\alpha}{2}}-1}\left(\sqrt{V_0}+\sqrt{C}\|\mathbf{z}_0\|\right) + 2^{-t}\cdot(2\rho D_3\|\mathbf{z}_0\|) \\
&\leq 24M\rho\sqrt{\frac{2m}{\mu}}\left(\sqrt{1-\frac{\alpha}{2}}\right)^t\left(\sqrt{V_0}+\sqrt{C}\|\mathbf{z}_0\|\right) + \left(\sqrt{1-\frac{\alpha}{2}}\right)^t\cdot(2\rho D_3\|\mathbf{z}_0\|) \\
&\leq \left(\sqrt{1-\frac{\alpha}{2}}\right)^t\left(24M\sqrt{\frac{2m}{\mu}}\left(\sqrt{V_0}+\sqrt{C}\|\mathbf{z}_0\|\right) + 2\rho D_3\|\mathbf{z}_0\|\right) \\
&\leq \rho\cdot D_4\left(\sqrt{1-\frac{\alpha}{2}}\right)^t\left(\sqrt{V_0}+\sqrt{C}\|\mathbf{z}_0\|\right),
\end{aligned}
$$

where $D_4 = 24M\sqrt{\frac{2m}{\mu}} + 2D_3$, and the first inequality is from splitting $\|\mathbf{A}\|_2^t = \|\mathbf{A}\|_2^{t-1}\cdot\|\mathbf{A}\|_2 \leq \left(\frac{1}{2}\right)^{t-1}\cdot\rho D_3$.

Therefore, we can obtain that

$$
\|\mathbf{z}_t\|^2 \leq 2\rho\cdot D_4^2\left(1-\frac{\alpha}{2}\right)^t\left(V_0 + C\|\mathbf{z}_0\|^2\right).
$$

Combining with the induction hypothesis that $V_t \leq \left(1-\frac{\alpha}{2}\right)^t\left(V_0 + C\|\mathbf{z}_0\|^2\right)$, we have

$$
\begin{aligned}
\sqrt{V_t}\cdot\|\mathbf{z}_t\| &\leq \rho D_4\left(1-\frac{\alpha}{2}\right)^t\sqrt{V_0 + C\|\mathbf{z}_0\|^2}\cdot\left(\sqrt{V_0}+\sqrt{C}\|\mathbf{z}_0\|\right) \\
&\leq 2\rho D_4\left(1-\frac{\alpha}{2}\right)^t\left(V_0 + C\|\mathbf{z}_0\|^2\right).
\end{aligned}
$$

Using Eqn. (8), we can get if $\rho < \frac{\alpha}{2\left(D_1 D_4 + D_2 D_4^2\right)}$,

$$
\begin{aligned}
V_{t+1} \leq & (1-\alpha)V_t + D_1 \sqrt{V_t} \cdot \|\mathbf{z}_t\| + D_2 \|\mathbf{z}_t\|^2 \\
\leq & (1-\alpha)\left(1-\frac{\alpha}{2}\right)^t \left(V_0 + C\|\mathbf{z}_0\|^2\right) \\
& + \rho \cdot \left(D_1 D_4 + D_2 D_4^2\right)\left(1-\frac{\alpha}{2}\right)^t \left(V_0 + C\|\mathbf{z}_0\|^2\right) \\
\leq & \left(1-\frac{\alpha}{2}\right)^{t+1} \left(V_0 + C\|\mathbf{z}_0\|^2\right).
\end{aligned}
$$

$\square$