[Reviews · NeurIPS 2020]

Review 1

Summary and Contributions: This paper presents DAPG, a novel decentralized proximal algorithm named, to solve the composite convex problem. The result shows that DAPG can achieve optimal computation complexity and near-optimal communication complexity. The authors convince me that DAPG is the first decentralized algorithm can achieve the optimal computation complexity and a near optimal communication complexity for the composite convex problem. Furthermore, this paper shows DAPG does not require each individual function to convex to achieve a linear convergence rate. This paper also validates the computation and communication efficiency of DAPG empirically and demonstrates DAPG outperforms state-of-art decentralized proximal algorithms. [response to rebuttal] After the discussion and reviewing the rebuttal, I insist on my score since the rebuttal and discussion convince me that the novelty is nontrivial and it is more a theory paper rather than experimental paper.

Strengths: A decentralized composite optimization problem has been widely studied. Before the work [1], decentralized proximal algorithms such as [17], [8] can only obtain sub-linear convergence rates. DPA [1] is the first decentralized proximal algorithm achieving a linear convergence rate. Then [24] proposes a framework to design decentralized proximal algorithms which can obtain linear convergence rates. However, the computation and communication complexities of previous works are sub-optimal. DPAG proposed in this paper is a novel decentralized proximal algorithm which achieves the optimal computation complexity and near-optimal communication complexity. Furthermore, DAPG does not require each $f_i(x)$ to be convex to achieve a linear convergence rate while this condition is required by previous works. These results are new and make the theory of decentralized proximal optimization more clear. DAPG uses a novel combination of gradient-tracking, multi-consense, and Nesterov's acceleration on the proximal algorithm. The theoretical analysis of DAPG is also novel. Because it is unclear to accelerated decentralized algorithms in an effective manner, I think the techniques used in DAPG would lead to a number of future works in developing accelerated decentralized proximal algorithms. Experiments also validate the computation and communication efficiency of DAPG. Decentralized optimization is an important research area in machine learning. Thus, I think this paper is closely relevant to NeurIPS.

Weaknesses: PG-EXTRA [17] and NIDS [8] can be used to the case that each agent has its own $r_i(x)$ and achieve a sub-linear convergence rate. Can DPAG be extended to this case and achieve a linear convergence rate? Furthermore, in Figure 1 and 2, I could not distinguish the results of DPA and PG-EXTRA. Though these two algorithms have similar convergence rates, it is better to use different colors and line styles to distinguish them.

Correctness: The claims of this paper are correct. The paper provides detailed proofs to prove the claims. Experiments also validate theoretical claims.

Clarity: The paper is well-written and easy to read.

Relation to Prior Work: This paper provides a comprehensive survey on the decentralized algorithm, including very recent works.

Reproducibility: Yes

Additional Feedback:


Review 2

Summary and Contributions: This paper develops a proximal version of the Mudag algorithm [25], (primal accelerated decentralized) to handle non-smooth regularizers. Therefore, it mimics centralized accelerated proximal gradient descent by performing approximate averaging using fast consensus after each update. The computation complexity is the same as accelerated proximal gradient descent, and the communication complexity is only multiplied by log factors that do not depend on epsilon (the final error). Avoiding this extra dependence on epsilon is permitted by the averaging + gradient tracking combination introduced by Mudag, which allows to warm-start the consensus error part. In the end, I believe that the contribution is rather incremental (Mudag for composite objective) but also nice to have.

Strengths: - Accelerated primal decentralized algorithm for composite minimization. - Well-written and clear. - Simple but convincing experimental section (datasets are not huge but that's ok for me).

Weaknesses: The main results of the paper are an incremental contribution over the results from [25]. Although it is great to have an algorithm that handles the composite setting, I am not quite sure that this is a strong enough result for publication at NeurIPS, especially since it does not appear to have required overcoming a key technical challenge. In particular: The Lyapunov function is the same on both papers Lemma 1 is an adaptation of Lemma 3 [25] Lemma 3 is an adaptation of Lemma 4 [25] Lemma 7 is an adaptation of Lemma 7 [25] Lemma 8 is the same on both papers It seems that the proof is mainly similar to Mudag's proof, adapting to the proximal setting with the usual tools (e.g., non-expansiveness of the proximal operator), where consensus after the prox step is ensured by performing an extra FastMix step.

Correctness: I have checked most of the proofs and I believe they are correct, though it is possible that I have missed something.

Clarity: Yes

Relation to Prior Work: Yes

Reproducibility: Yes

Additional Feedback: === edit === I have read the rebuttal, and I would have liked the authors to point out precisely *what* technical points change and are difficult to handle. I think it would be great to actually highlight them in a revision of the paper. On a side note, I still believe that it is possible to get rid of the consensus step on y_t, and closeness between y_t and \bar{y}_t should be enforceable by the consensus step on x_t. This should be better in practice, since the consensus steps that are currently performed on y_t would also benefit x_t. A comparison with higher values of K would also have been welcome. In particular, does the algorithm still converge (though at a slower rate) when K=1 and the eigengap of the graph is very small? In the end, the rebuttal partially adressed my concerns but I keep my original score. ========== What are the key differences in terms of proofs? Does the decentralized aspect come into play or is it just going through the Mudag proof and adapting to the proximal setting with the usual tools (e.g., non-expansiveness of the proximal operator), where consensus after the prox step is ensured by performing an extra FastMix step? I believe that the authors should insist more on the differences with Mudag, and how it is interesting to extend it to the prox setting (in particular insisting on the technical difficulties, if any) in order to highlight the technical contribution. Why perform an extra FastMix step on y_{t+1}? Isn't it better to perform more steps on x_{t+1} (and therefore on x_t before that) if necessary, since y_{t+1} is a linear combination of x_{t+1} and x_t (and so preserves the consensus)? I get why 1 extra consensus step is needed over Mudag but not 2. Typos: l80: if and if only => if and only if l102: difference => different L141: following lemma show*s* / following lemma*s* show


Review 3

Summary and Contributions: The paper introduces a new algorithm for solving strongly convex decentralized optimization problems. The algorithm is based on accelerated proximal primal update, which is compatible with non-smooth regularization, achieving optimal computational and communication complexity (up to logarithmic factors). Experiments are conducted showing the effectiveness of the algorithm.

Strengths: The paper introduces a theoretical grounded decentralized optimization algorithm to address non-smooth regularization. The main idea is to alternatively apply a proximal gradient type algorithm, and, a multi-consensus step that propagates the local update. Convergence analysis are provided based on Lyaponov functions, showing that the proposed algorithm achieves optimal computational and communication complexity (up to logarithmic factors). The paper is theoretically solid.

Weaknesses: My main concern of the paper is about the novelty of the algorithm. Indeed, the proposed algorithm can be viewed as a proximal variant of the Mudag algorithm [1]. The key components such as gradient tracking and multi consensus are already introduced in Mudag and even earlier, which leaves the only contribution to be incorporating the proximal operator. Moreover, the proof is based on a similar Lyaponov function as in Mudag, which makes the contribution limited. Furthermore, the experimental section is not very convincing to me due to the choice of communication parameter K. This parameter controls the number of communication rounds after each proximal update, which is essential to achieve the optimal convergence rate. In theory, this parameter is proportional to the condition number (taking square root); in practice, it is set to be 1, which is significantly smaller compared to the theoretical choice. I understand that one would prefer to take K as small as possible to reduce the communication round, which is advantageous for the proposed algorithm. However, the well behavior with a small K may suggest that the graph is well connected, in which case one may not need to communicate too often. I believe a study with different graph structure can help understanding this phenomenon better, for example taking a ill conditioned graph such as a cycle. Moreover, another way to make the problem harder is to distribute the data in a non i.i.d manner, for example forcing half of the agents have access to positive labels and the other half to negative labels. In this case, communication are necessary and a different regime of the parameter K might occur. [1] Ye, H., Luo, L., Zhou, Z., & Zhang, T. (2020). Multi-consensus decentralized accelerated gradient descent.

Correctness: I have quickly gone through the proof, the theoretical results are correct to the best of my knowledge.

Clarity: The paper is written in a technical way where more intuition will be desirable.

Relation to Prior Work: A more detailed comparison with Mudag is missing, for example in the second bullet in page two of the contribution, it states that the algorithm does not requires each individual function to be strongly convex, is Mudag also satisfied this property? I believe the algorithm APM-C [2] is also related to the discussion since it applies multi-consensus step and it is also compatible with non smooth regularizers. [2] Li, H., Fang, C., Yin, W., & Lin, Z. (2018). A sharp convergence rate analysis for distributed accelerated gradient methods

Reproducibility: Yes

Additional Feedback: == EDIT after author's response == I increase my score given that this is the first algorithm to achieve accelerated (optimal) convergence rate for non smooth decentralized problems. Even though I still think the main credit should be given to Mudag algorithm, the proposed algorithm do have merit since combining the proximal operator is not always trivial (the proposed method has introduced an additional consensus step).


Review 4

Summary and Contributions: This submission deals with decentralized composite optimization by proposing an accelerated proximal algorithm that achieves near optimal computational and communication complexity. ==== After authors' response === I have read the rebuttal and increased my score. Despite my concerns on novelty is partially addressed, the original assessment that the presentation can benefit from improved writing still stands. In addition, the numerical tests should be also expanded.

Strengths: i) An accelerated proximal algorithm is developed with near-optimal computational and communication complexity. ii) Local (strong) convexity is not necessary for the established bound.

Weaknesses: i) The main concern is about novelty of this work. Because it is closely related to the works in [9, 25], a detailed comparison is certainly needed. The main algorithm is not clearly described so that the intuition and the idea behind it, are not easy to decipher. For example, what will happen if all the FastMix steps in Alg.1 are replaced by standard (K rounds of) communication? ii) It is unclear what will happen f_i(x) is \nv strongly convex? iii) The proposed **accelerated** algorithm is mostly compared with those non-accelerated ones in the remarks. Additional elaboration is due to compare with other decentralized **accelerated** algorithms. iv) The notation in (11), namely F(x), is not consistent with that of (1). v) The numerical tests only compare the proposed accelerated method to those non-accelerated ones, which is not fair. Consider comparsions with accelerated ones, e.g., [14]. In addition, the parameter choice K=1 seems to be very different from what the theoretical analyses suggest.

Correctness: I went over the proof, and it seems to be correct. As mentioned earlier, the concerns pertain to empirical methodology.

Clarity: There is a need to elaborate on the intuition behind the main algorithm (Alg. 1). While the established bounds are explained, they are only compared with those non-accelerated ones.

Relation to Prior Work: See earlier discussion on this issue.

Reproducibility: Yes

Additional Feedback:

[Author Response · NeurIPS 2020]

We appreciate the valuable comments from the reviewers. We will revise them accordingly.

**Novelty of DAPG and Brief History of Decentralized Proximal Gradient Descent**

The novelty of DAPG is the main concern of reviewers because DAPG is closely related to Mudag [25]. However,
reviewers don't know and we don't emphasize that it is hard to extend decentralized gradient descent methods with
gradient tracking to proximal counterpart with the same linear convergence rates. Please note that, researchers take *five*
*years* to propose DPA [1], which is the first linearly convergent decentralized proximal gradient algorithm, after the
publication of EXTRA [19], the first decentralized gradient descent method of linear convergence rate.

PG-EXTRA [17], the follow-up work of EXTRA, extends EXTRA to the composite setting which has extra non-smooth
term, but only obtains the sub-linear convergence rate even for strongly convex $f_i(x)$. In the following years, different
decentralized gradient descent algorithms with gradient tracking were proposed such as [13], [14]. However, no
evidence shows that these algorithms can be extended to the composite setting but keeping linear convergence. Even the
recent work NIDS [8] can only achieve a sub-linear convergence rate for the composite setting. In fact, until five years
after EXTRA, DPA [1] is the first linearly convergent proximal gradient algorithm for decentralized optimization which
is published in Neurips 2019. Just as the title of [13]-'Harnessing smoothness to accelerate distributed optimization'
mentioned, gradient tracking tries to harness the smoothness of the function. However, due to non-smooth term, the
convergence analysis of gradient tracking based methods become much harder.

From above history, we can observe that, in the decentralized setting, extending the results of decentralized gradient
descent methods to their proximal counterparts is not an easy work. Thus, whether Mudag [25] can be extended to the
decentralized proximal setting is not obvious. In fact, just as pointed by Reviewer 2, to deal with the non-smoothness,
DAPG takes more consensus steps compared to Mudag. The proof of Lemma 2 is also totally different from the one of
Lemma 9 of Mudag. Note that, the proof of DAPG does not rely on the technique of DPA, either.

Furthermore, the results obtained by DAPG are total new for decentralized proximal algorithms. To the best of
our knowledge, DAPG is the first accelerated decentralized gradient descent which theoretically outperforms current
decentralized proximal algorithms.

Thus, the novelty and contribution of DAPG are substantial.

**Reviewer_1** : Thank you for the comments on figures, we will revise accordingly.

**Reviewer_2**

Q1:Incremental contribution to Mudag

A1: Just as mentioned at the beginning, it is not easy to extend decentralized gradient descent with gradient tracking
to the composite setting. Because of the non-smoothness of composite setting, DAPG has more consensus steps than
Mudag and the proof of Lemma 2 is totally different from the one of Lemma 9 of Mudag.

**Reviewer_3**

Q1: Experiment and Relation to prior work and Reference to APM-C

A1: We should set $K$ according to the condition number of graph. We will give more experiments on graphs of large
condition numbers in our revised paper. We will give more detailed comparison with Mudag and will add the reference
of APM-C in our revised paper.

**Reviewer_4**

Q1: Comparison with Accelerated version of decentralized proximal gradient descent methods?

A1: As mentioned at the beginning, DPA [1] is the first decentralized proximal gradient descent method with linear
convergence rate. NIDS is shown to achieve linear convergence rate in [24] which is the best convergence rate of
decentralized proximal gradient methods can achieve before our work. Work [14] is a decentralized accelerated gradient
descent method which can not deal with the non-smooth regularization term and no evidence shows that it can be
directly extended to deal with the non-smooth regularization term. To the best of our knowledge, DAPG is the first
decentralized accelerated proximal gradient descent method. Thus, we have compared DAPG with state-of-the-art
algorithms theoretically and empirically.

Q2: What will happen if all the FastMix steps in Alg.1 are replaced by standard (K rounds of) communication? It is
unclear what will happen $f_i(x)$ is $\nu$ strongly convex?

A2: By standard communication, the communication complexity of algorithm will depend on $1/(1 - \lambda_2(W))$ instead
of $1/\sqrt{1 - \lambda_2(W)}$. DAPG does not require $f_i(x)$ to be $\nu$ strongly convex. But if $f_i(x)$ is $\nu$ strongly convex, the results
of Theorem 1 of course still hold.

[Meta-Review · NeurIPS 2020]

The paper gives an accelerated gradient method for decentralized optimization on composite objectives. It achieves this by mimicking centralized accelerated proximal gradient descent. Slight concerns remained about the level of novelty over the Mudag algorithm, which should be expanded in the discussion more precisely, as well as the (theory) requirement of K>1 communications after every step and the not yet fully explained dependence of K on the graph parameter. We expect the authors to incorporate the feedback and improvement suggestions from the 4 reviews in the camera ready version.